# Relationships between community composition, productivity and invasion resistance in semi-natural bacterial microcosms

**Matt Lloyd Jones[†]*, Damian William Rivett[‡], Alberto Pascual-García[§], Thomas Bell**

Department of Life Sciences, Imperial College London, Silwood Park Campus, Ascot, United Kingdom

**\*For correspondence:**
matt.lloydjones@outlook.com

**Present address:** [†]Environment and Sustainability Institute, University of Exeter, Penryn, Cornwall, United Kingdom; [‡]Department of Natural Sciences, Faculty of Science and Engineering, Manchester Metropolitan University, Manchester, United Kingdom; [§]Institute of Integrative Biology, ETH Zürich, Zürich, Switzerland

**Competing interests:** The authors declare that no competing interests exist.

**Abstract** Common garden experiments that inoculate a standardised growth medium with synthetic microbial communities (i.e. constructed from individual isolates or using dilution cultures) suggest that the ability of the community to resist invasions by additional microbial taxa can be predicted by the overall community productivity (broadly defined as cumulative cell density and/or growth rate). However, to the best of our knowledge, no common garden study has yet investigated the relationship between microbial community composition and invasion resistance in microcosms whose compositional differences reflect natural, rather than laboratory-designed, variation. We conducted experimental invasions of two bacterial strains (*Pseudomonas fluorescens* and *Pseudomonas putida*) into laboratory microcosms inoculated with 680 different mixtures of bacteria derived from naturally occurring microbial communities collected in the field. Using 16S rRNA gene amplicon sequencing to characterise microcosm starting composition, and high-throughput assays of community phenotypes including productivity and invader survival, we determined that productivity is a key predictor of invasion resistance in natural microbial communities, substantially mediating the effect of composition on invasion resistance. The results suggest that similar general principles govern invasion in artificial and natural communities, and that factors affecting resident community productivity should be a focal point for future microbial invasion experiments.

## Introduction

Microbial communities of all types are challenged by the arrival of dispersing microbes, which may displace resident taxa and alter ecosystem functioning (*Amalfitano et al., 2015*; *Fernandez-Gonzalez et al., 2021*; *Kinnunen et al., 2016*; *Litchman, 2010*; *Mallon et al., 2015a*; *Thakur et al., 2019*). A primary way in which established communities defend against such invasion attempts is by achieving a high level of productivity (strictly defined as community members' growth rate but often approximated by standing biomass) in their home environment before invaders arrive (*Huston, 2004*). High levels of productivity are concomitant with the depletion of resources, the establishment large populations and the occupation of physical space – and consequently, invaders' (in) ability to survive in that environment (*Ghoul and Mitri, 2016*; *Huston, 2004*; *Stubbendieck et al., 2016*).

Studies with simplified, synthetic microbial communities have suggested that community productivity is of such importance to a microbial community's invasion resistance that it is often the main explanation for the effect that the composition of the community inoculated into the microcosm has on invasion resistance (*Eisenhauer et al., 2013*; *Hodgson et al., 2002*; *van Elsas et al., 2012*; *Yang et al., 2018*). In these studies, microcosms containing a sterile culture medium (e.g. lab broth,

**eLife digest** Much like animals and plants, microorganisms such as bacteria and fungi naturally live in communities, where different species exist together and share the same resources. These communities can be quite stable over time and resist the invasion of new species – for example, by collectively and rapidly consuming all the available resources before invaders arrive. The gut microbiome is one example of such a microbial community, but there are many others.

There have been many studies of how artificial microbial communities created in the lab resist invasion, but it remains unclear how naturally-occurring microbial communities do so, because they are harder to study in the lab. A leading theory is that certain combinations of microbes (i.e. communities) grow and consume resources faster than other combinations – this is known as achieving high productivity.

Jones et al. conducted invasion experiments across hundreds of naturally-occurring microbial communities collected from woodland puddles that form in the exposed roots of beech trees. Each community contained different combinations of bacteria, but they all largely survived by breaking down leaf litter, so Jones et al. created a tea from beech leaves in which to grow these natural communities in the lab. The relationships between community composition, productivity and invasion resistance were then assessed using a combination of DNA sequencing, measurements of community growth and measurements of invader survival. Jones et al. found that natural combinations of bacteria that grew well together drove invasion resistance in these communities, mirroring results seen in much more artificial communities grown in the lab.

These results suggest that productivity is a key factor underpinning invasion resistance in naturally-occurring microbial communities. This is a useful insight that could shape thinking about how the long-term stability of beneficial microbial communities – such as healthy gut microbiomes – might be improved, and how harmful communities – such as dental plaques – could be destabilised. The next step will be to conduct similar experiments in other natural microbe communities to see how generally applicable these results are.

autoclaved soil) are inoculated with artificial microbial communities constructed from different combinations of culturable strains and/or dilutions of natural communities (typically 90% at each dilutions step). Each of these communities – differing in their composition – is then invaded with the same population of microbes to assess the relationship between community composition, productivity and invasion resistance. Such experiments have shown that the presence of individual and/or combinations of resident species (composition) with the highest cell densities and/or growth rates (productivity) explain most of the variation in invasibility between communities (*Eisenhauer et al., 2013*; *Hodgson et al., 2002*; *van Elsas et al., 2012*; *Yang et al., 2018*). Furthermore, since higher diversity communities are generally more productive (*Bell et al., 2005b*), there is often a negative relationship between a resident community's diversity and invasibility (e.g. *Hodgson et al., 2002*).

However, the relationship between a microbial community's composition, productivity and its invasion resistance is far from clear-cut. For example, *Hodgson et al., 2002* noted that whilst the effects of composition on invasion resistance mainly manifested as productivity differences, there was also evidence of productivity-independent effects of composition (e.g. niche complementarity, antagonism, facilitation). Similarly, *De Roy et al., 2013* found no clear relationship between composition and productivity in artificial microbial communities grown in complex media, but nonetheless a clear, strong relationship between resident community composition and invasion resistance. These results from experiments with artificial communities suggest that despite the predominance of productivity, there is scope for productivity-independent effects of composition on invasion resistance. Such direct effects of composition might become more apparent in natural communities, where niche space is likely to be more highly packed and productivity differences less pronounced (*Eisenhauer et al., 2012*). However, the relationships between microbial community composition and invasion resistance have not been studied in communities whose compositional differences reflect naturally occurring (rather than lab-engineered) compositional differences.

Here, we use natural microbial communities to test the hypotheses that (A) resident community productivity is a primary factor underpinning invasion resistance and (B) the effect of resident community composition on invasion resistance is mostly mediated by productivity. We conducted experimental invasions into 680 bacterial communities collected from rainwater pools (water-filled beech tree holes/phytotelmata). These types of model communities have been extensively used in previous microcosm experiments (*Bell et al., 2010*; *Bell et al., 2005b*; *Fiegna et al., 2015a*; *Fiegna et al., 2015b*; *Foster and Bell, 2012*; *Glücksman et al., 2010*; *Jones et al., 2017*; *Lawrence et al., 2012*; *Rivett et al., 2016*; *Rivett and Bell, 2018*; *Scheuerl et al., 2020*) and are particularly useful for conducting biodiversity-ecosystem functioning type experiments because diversity is likely to be determined largely by habitat size in nature (*Bell et al., 2005a*; *Woodcock et al., 2007*).

We grew and invaded these communities in a common garden, complex medium reflecting their natural growth medium (a beech leaf-based 'tea'), assessing the survival of two invaders (*Pseudomonas fluorescens* and *Pseudomonas putida*) at 24, 96, and 168 hr (1, 4, and 7 days) post-invasion (*Figure 1A*). Such invasion conditions are similar to that used by *van Elsas et al., 2012*, in which an established community was inoculated with an approximately equal density of invading cells. This scenario is probably relatively common in microbial communities (*Mallon et al., 2018*; *Rillig et al., 2015*) where an established community may frequently face the invasion of an entire microbial community (e.g. during faecal deposition on soil, leaf fall into soil or water, flooding of terrestrial ecosystems by aquatic ecosystems). For example, as noted by *Mallon et al., 2018*, deposition of faeces into soil may introduce as much as $10^9$ *E. coli* cells per gram (*Tenaillon et al., 2010*). For our tree hole system in which autumn leaf fall is a main input (beech tree holes were often found to be full or overflowing with leaves during fieldwork), there is evidence that in nature there are $10^6$–$10^9$ bacteria per gram of beech leaves (*Holm and Jensen, 1972*) falling into phytotelma containing approximately $10^6$ bacteria per ml (*Walker et al., 1991*). Thus, we believe our conditions are a somewhat realistic representation of the high-magnitude, low-frequency autumn invasion of bacteria into tree hole microbial communities that have established over the previous year.

Following previous experiments, we analysed the relationship between invader survival and the composition of the community added to each microcosm at the start of the experiment. Unlike previous experiments using synthetic microbial assemblages, the inoculated communities were naturally occurring bacterial assemblages. We quantified the starting community composition of the microcosms using amplicon sequencing (16S rRNA locus) to estimate the sequence abundances of OTUs inoculated into the microcosms (Day 0). As well as estimating the starting community composition in this way (i.e. 16S genotypes present), we also took various phenotypic measurements of the communities at 7 and 14 days of growth in the laboratory microcosms (the latter being the day of invasion). Community productivity was the primary community-level phenotype of interest to us, quantified by measuring cell density and respiration before invasion. Additionally, we took phenotypic measurements related to metabolic activity as an alternative hypothesis to productivity, and to further contextualise any results related to composition and productivity. These measurements were the potential metabolic activity (ATP levels) and capacity of the communities to degrade a set of four specific substrates that we expected to be important components of these environments (cellulose, chitin, xylose, and phosphate). The experiment allowed us to identify the main components of invasion resistance in natural communities, and assess whether the effects of resident community composition on invasion resistance primarily manifest as effects on productivity.

## Results

### Most important explanatory variables for invader survival

To select the components of resident community starting composition and realised community phenotype that best predicted the two invaders' survival in the microcosms across the three sampling points, we compared explanatory variable importance using random forest regressions (Materials and methods: Statistical techniques). Random forest regressions included all explanatory variables relating to starting composition and realised phenotype. The best representation of composition was selected by testing different dimensionality reduction techniques and selecting the best compromise between variance explained and number of variables. We selected a network approach

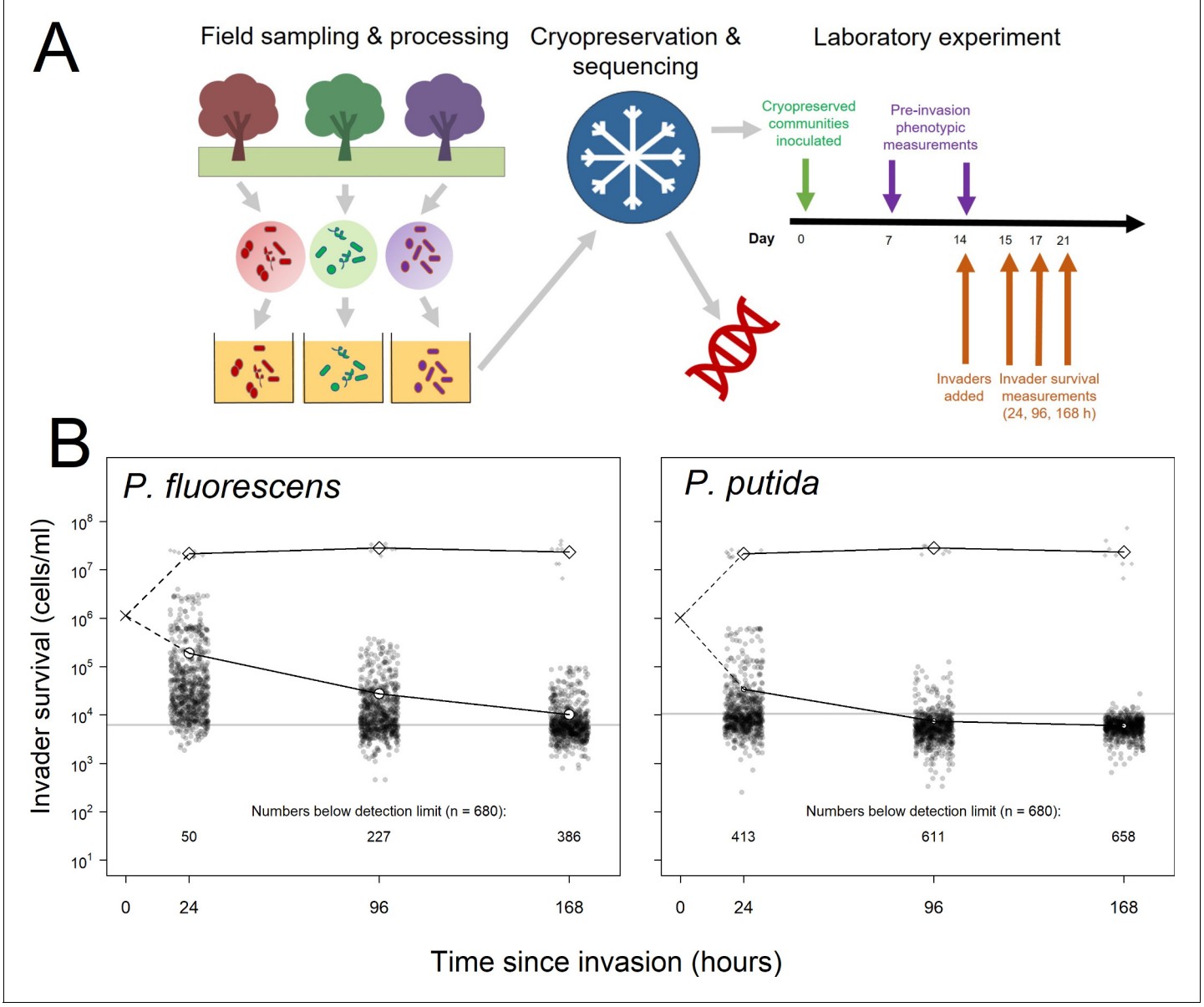

**Figure 1.** Summary of experimental set-up and broad patterns of invader survival across the three sampling points. (**A**) Schematic depicting the sampling and processing of communities (field sampling and growth of lab acclimation of communities), the cryopreservation and sequencing of the lab-acclimated communities, and the setup and sampling scheme of the laboratory experiment described here. (**B**) Invader survival values for both invaders at each of the three sampling points in monoculture (diamonds) and in communities (circles). Larger, white points represent the means for the respective subsets of the data; grey line represents the estimated cells/ml detection limit; dashed line represent inferred trajectories between the inoculation density and the invasion densities, as the inoculation density was measured in the invader culture prior to its inoculation into communities. The online version of this article includes the following source data for figure 1:

**Source data 1.** Invader survival data for each of the 680 communities after averaging across the four pseudoreplicated assays and converting from lux to cells/ml.

**Source data 2.** Data from the growth-curve assay of luminescence and plate count measurements, used to calibrate invader luminescence against cell density.

**Source data 3.** Table in the same format as as *Figure 1—source data 1* but with TRUE/FALSE values instead of values indicating which invader survival measurements were below the detection limit of 12 lumens (TRUE) before conversion to cells/ml.

reducing composition to key species' communities, hereafter termed functional groups (see Materials and methods: Computational techniques).

Explanatory power of the random forest regressions ranged from 8.14% to 59.37% (pseudo $R^2$), with substantially better performance predicting invasions at 24 compared to 96 and 168 hr, and for *P. fluorescens* compared to *P. putida* invasions. This correlated with the number of invasions falling below the detection limit of the invasion assays; invader survival declined with time since invasion, and was also lower for *P. putida* than for *P. fluorescens* (*Figure 1B*; Materials and methods: Laboratory techniques). Note that the random forest method somewhat accounts for the difficulty of differentiating noise from true signal below the detection limit by iteratively sub-sampling which samples are included in each of its constituent regression trees. However, where there are larger numbers of invasions falling below the detection limit, this becomes more difficult and hence explanatory power drops. Nonetheless the method was sufficient to identify the most important explanatory variables for downstream analysis, where we account for the detection limit problem with a sensitivity analysis (Materials and methods: Statistical techniques).

Across random forests, phenotypic measures related to community productivity before invasion were by far the strongest and most consistent individual predictors of invader survival (*Figure 2*). Measures of community cell yield and respiration consistently had the highest variable importance values – quantified as the relative increase in the Mean Square Error (% IncMSE) obtained when the data associated to the variable under analysis is absent from a regression (*Figure 1*). The variable importance of cell yield and respiration was generally higher than that of other variables relating to either phenotype or the starting composition of the microcosms. However, the abundance of Functional Group 18 (containing 12 OTUs assigned to *Cedecea spp.*, *Citrobacter werkmanii*, *Erwinia persicina*, *Erwinia rhapontici*, *Escherichia shigella spp.*, *Klebsiella pneumoniae*, *Pantoea agglomerans*, *Pantoea vagens*, *Serratia fonticola*, *Serratia liquefaciens*, *Serratia quinivorans*, and *Trabulsiella spp.*) was sometimes of comparable variable importance to the productivity variables. All these variables were approximately linearly and negatively correlated with invader survival; increasing cell yield, respiration and abundance of Functional Group 18 was correlated with lower invader success (*Figure 3*, *Figure 3—figure supplement 1*).

Other variables relating to starting composition were weaker individual predictors of invader survival. As well as Functional Group 18, other functional groups including Functional Group 5 (*Acinetobacter genomospecies 3*, *Acinetobacter towneri*, *Novispirillum itersonii* and *Ralstonia pickettii*) and 20 (*Acidovorax spp.*, *Acinetobacter calcoaceticus*, *Acinetobacter johnsonii*, *Aquabacterium spp.*, *Brevundimonas aurantiaca*, *Caenimonas spp.*, *Delftia lacustris*, *Herbaspirillum rubrisubalbicans*, *Herbaspirillum spp.*, *Leptothrix spp.*, *Massilia timonae*, *Paucimonas spp.*, *Phenylobacterium spp*, *Pseudomonas balearica*, *Pseudomonas pseudoalcaligenes* and *Stenotrophomonas maltophilia*) also appeared to be somewhat important for invader survival. There was some evidence of a weak negative relationship between Simpson's diversity and invader survival (*Figures 2,3 Figure 3—figure supplement 1*) – although the abundance of key functional groups was a more reliable indicator of invader survival across time-points and invaders (*Figure 2*). There was little evidence for a strong effect of phylogenetic diversity (Rao's quadratic entropy; a phylogenetic equivalent to Simpson's diversity index) or phylogenetic distance of the community from the invader (although these results are subject to the reliability of 16S phylogenetic tree; see Discussion: Unexplained variation).

Overlaying the abundance of the most important functional group (Functional Group 18) on the above described relationships (*Figure 3*, *Figure 3—figure supplement 1*) suggested that the presence of particular 16S genotypes determined overall community phenotype (in terms of productivity and invasion resistance realised in laboratory microcosms). Highly invadable communities with a low cell density and respiration were also those with a low abundance of Functional Group 18 OTUs and similarly, communities with a low abundance of Functional Group 18 had a lower level of respiration (blue colours in *Figure 3* and *Figure 3—figure supplement 1*).

There was little evidence for strong effects of more specific measures of community phenotype, with ATP activity and the potential to degrade specific substrates having a weak effect overall (*Figure 2*; *Figure 3—figure supplements 2* and *3*).

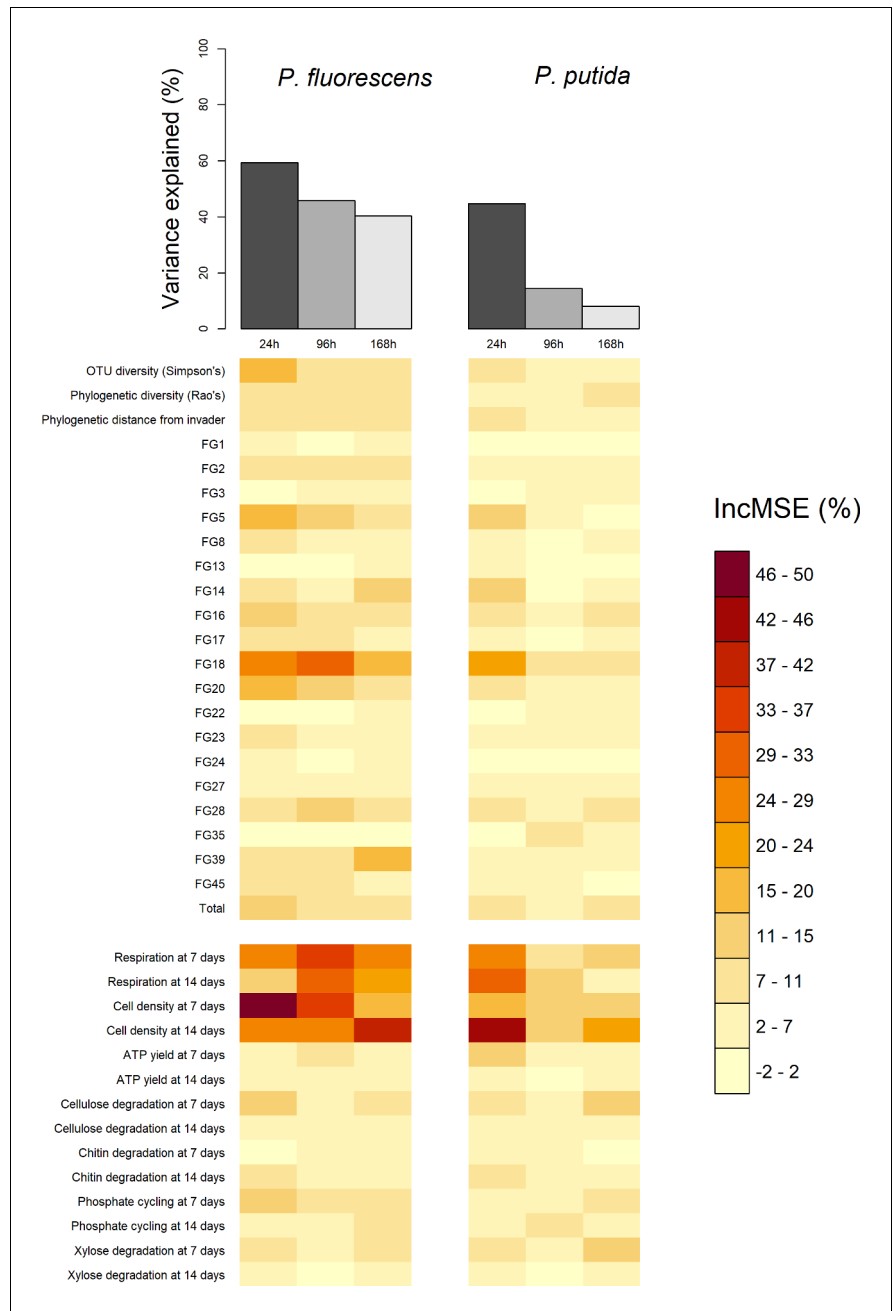

**Figure 2.** Comparison of total variance explained (top bars) and the variable importance values (bottom heatmap) of the six random forests, computed for each of the two invaders and each of three the invader survival sampling points at 24, 96, and 168 hr post-invasion. Total variance explained is calculated as pseudo R-squared: 1-Mean Squared Error/variance (invader survival) of the random forest. Variable importance values are the percentage increase in Mean Squared Error (IncMSE %) when the variable is not permuted i.e. a high (low) value represents a variable of high (low) importance to explaining invasion success. Each column in the variable importance heatmap represents the variable importance values of the random forest using functional Groups represented by the orange bar in the top figure. The heatmap is split into compositional (above split) and functional (below split) variables. Compositional variables labelled 'FG+number' refer to the functional group ids.

The online version of this article includes the following source data and figure supplement(s) for figure 2:

**Source data 1.** Table of the variance explained by each random forest with key columns being invader (invader assayed – *P. fluorescens SBW25* or *P. putida KT2440*), timepoint (time since invasion), and varexp (pseudo R-squared/% variance explained).

*Figure 2 continued on next page*

*Figure 2 continued*

**Source data 2.** Invader survival data for each of the 680 communities after averaging across the four pseudoreplicated assays and converting from lux to cells/ml.
**Figure supplement 1.** Rank abundance plot of OTUs (mean and standard error of each OTU's abundance in all communities).
**Figure supplement 2.** First two coordinates (of five total used in the analysis) of the principal coordinates analysis (PCoA).
**Figure supplement 3.** Comparison of the performance of different dimensionality reductions of the starting composition data, shown as mean variance explained vs the number of dimensions in each of the tested reductions. The functional groups approach had a disproportionate explanatory power for its number of dimensions, comparable to no dimensionality reduction, and so we opted for this method for our main analyses.

## Mediation of the effect of starting composition by productivity (structural equation models)

Having identified community productivity as the most important predictor of invasion resistance and identified the abundance of functional groups inoculated into the microcosms as the putative cause of this, we aimed to estimate the extent to which the effect of starting composition on invasion resistance was mediated by productivity. We used structural equation models (SEMs) to identify whether starting community composition was associated with lower invader survival solely because it determined the productivity of communities. Having specified the latent variables using the most important explanatory variables (the functional group abundances and the two measures of cell density; see Materials and methods: Statistical techniques), we sought to understand the mediation of composition by productivity across our experiments by using three different SEMs in which these latent (dependent) variables Composition and Productivity influence the latent (independent) variable Invasion following one of these hypothesis:

1. No mediation – only resident community starting composition (directly) affects invaders' survival
2. Partial Mediation – resident community starting composition (directly) affects invaders' survival, but also has an indirect effect through its influence on community productivity, which in turn affects invader survival.
3. Complete Mediation – resident community starting composition only influences invaders' survival through its indirect effect on community productivity.

Model comparison strongly supported the idea that the effect of starting composition on invader survival was strongly mediated by realised productivity, with the No Mediation model being easily rejected ($\Delta$AIC = 166.98). The Partial Mediation model was the best fitting of the three models and was significantly better than the Complete model (likelihood ratio test, $\Delta$Chi$^2$ = 7.81, pval = 0.0052) - though this should be interpreted with caution as the quality of the models is not optimal (CFI ~ 0.63). Based on the estimated coefficients of the best, Partial Mediation model (*Figure 4*), we estimated that a minimum of 48% of the effect starting composition on invader survival (the product of coefficients C→P x P→I) was mediated by productivity (0.47 x −0.63 = −0.30; which represents a proportion −0.30/−0.63 = 0.48 of the total effect). Furthermore, comparing the total (direct + indirect) effects of productivity (P→I + C|P→I) and composition (C→I + C|P→I) revealed that the total effect of productivity (−0.33 + −0.3 = −0.63) was much stronger than that of composition (−0.13 + −0.30 = −0.43).

Given the relatively small (but significant) difference between the Partial and Complete mediation model fits, the strong effects of productivity in the Partial model, and the sub-optimal quality of the models, we conducted an additional sensitivity analysis (see Materials and methods: Statistical methods). This sensitivity analysis tested how sensitive model selection was to the invader survival values fell below the strict detection limit of the invasion assay (see Materials and methods: Laboratory techniques). The sensitivity analysis supported the Complete (the best model in 97.4% of permutations) over the Partial mediation model (the best model in 2.6% of permutations), highlighting the marginality of the model selection result.

Taking everything into account, we believe the most conservative interpretation of these results, therefore, is that most of the effects of composition occurred through productivity.

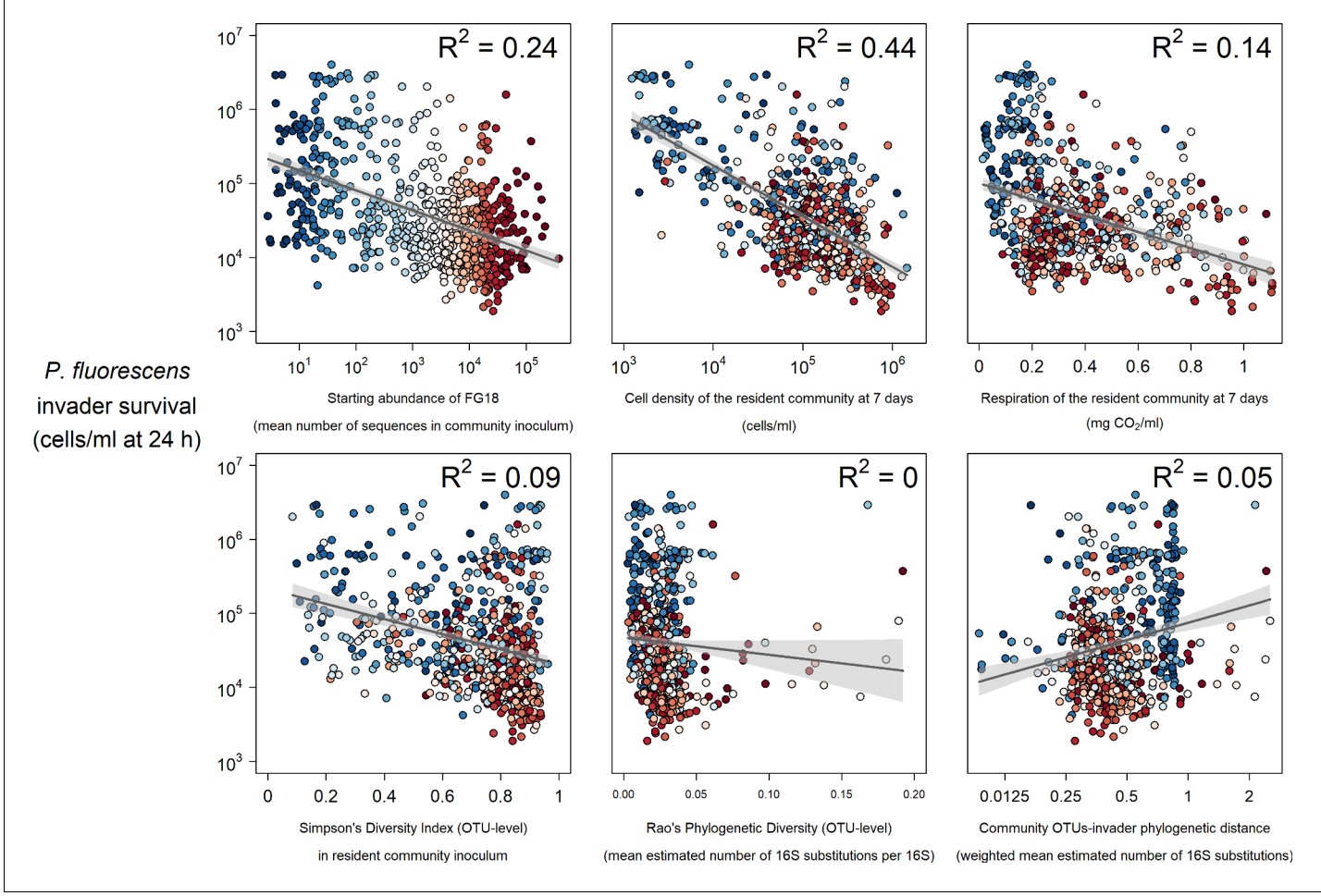

*P. fluorescens* invader survival (cells/ml at 24 h)

**Figure 3.** Selected strong (top) and weak (bottom) relationships between explanatory variables and *P. fluorescens* invasion success at 24 hr post-invasion. Colours represent the mean abundance of OTUs belong to Functional Group 18 in each community (blue low, red high).

The online version of this article includes the following source data and figure supplement(s) for figure 3:

**Source data 1.** Invader survival data for each of the 680 communities after averaging across the four pseudoreplicated assays and converting from lux to cells/ml.

**Source data 2.** Diversity metrics for each of the 680 communities (Simpson's diversity, Rao's quadratic entropy, and phylogenetic distance of the invader from the community for each of the two invaders).

**Source data 3.** Abundance of each of the functional groups in each of the 680 communities (mean number of sequences in the community inoculum for OTUs belonging to that group).

**Source data 4.** Data relating to the phenotypic assays performed/measurements taken at 7 and 14 days, before invasion at 14 days.

**Source data 5.** Diversity and phenotypic assay data, combined into one table for convenience.

**Figure supplement 1.** Selected strong (top) and weak (bottom) relationships between explanatory variables and *P. putida* invader survival at 24 hr post-invasion.

**Figure supplement 2.** Relationships between enzyme activity and *P. fluorescens* invader survival at 24 hr post-invasion.

**Figure supplement 3.** Relationships between enzyme activity and *P. putida* invader survival at 24 hr post-invasion.

## Discussion

We found that even in complex, species-rich bacterial communities, community productivity is a key determinant of invasion resistance, with much of the effect of community composition manifesting as effects on productivity. This is consistent with the results of earlier experiments with simpler, more artificial communities (*Bonanomi et al., 2014*; *Eisenhauer et al., 2013*; *Hodgson et al., 2002*; *van Elsas et al., 2012*). Productivity (pre-invasion cell density and to a lesser extent respiration) were certainly the most useful individual explanatory variables for invasion success in our experiments, though with certain compositional variables also having a strong effect (*Figure 2*). Furthermore,

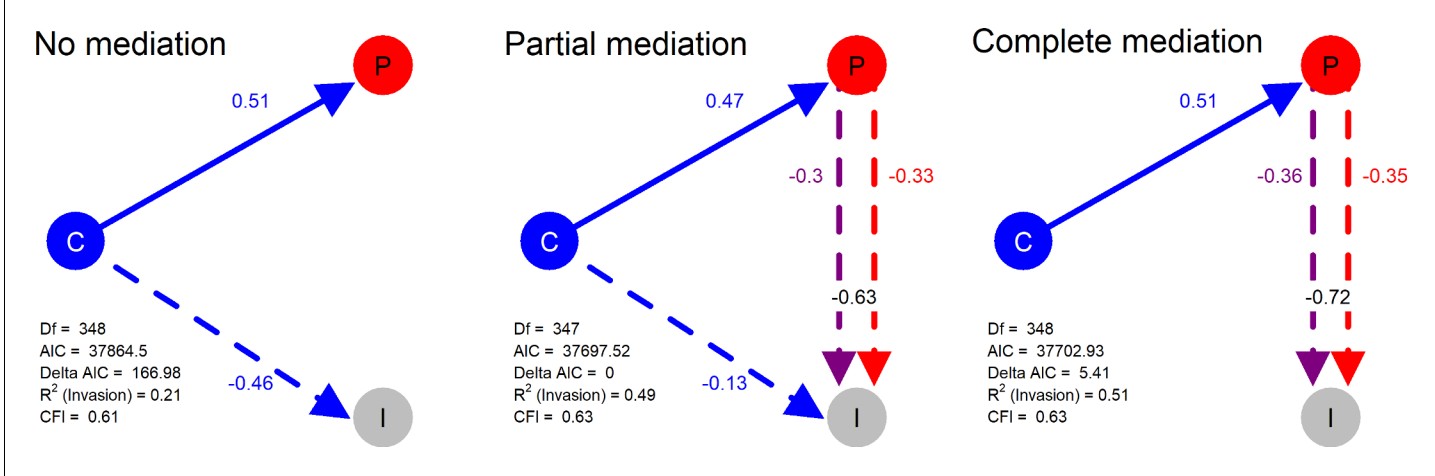

**Figure 4.** Structural models in a mediation test. Nodes in the diagrams represent latent variables C = Composition (blue), P = Productivity (red), I = Invasion (grey). Arrows between nodes represent regressions between dependent and independent variables (blue = direct effect of composition, red = direct effect of productivity, purple = composition effect mediated by productivity). Direction and value of each path is indicated by arrow type (positive effect = solid line, dotted line = negative effect) and the standardised regression coefficient adjacent to the arrow. In the path P → I the total effect of productivity (sum of direct and composition-mediated effects) is shown between both lines in black.

The online version of this article includes the following source data and figure supplement(s) for figure 4:

**Source data 1.** Model comparison results for the three structural equation models.
**Source data 2.** Model coefficients for the three structural equation models.
**Figure supplement 1.** Full structure of Partial Mediation structural equation model.
**Figure supplement 2.** Full structure of Complete Mediation structural equation model.
**Figure supplement 3.** Full structure of No Mediation structural equation model.

structural equation modelling suggested that much of compositional effect was due to the effect that the bacteria inoculated into that microcosms had on productivity, which in turn affected invasion success (*Figure 4*). Nonetheless, there is likely a still a (smaller) role for more direct, productivity-independent effects of composition – again, consistent with previous work with artificial communities (*De Roy et al., 2013*; *Hodgson et al., 2002*), and for composition-independent effects of productivity.

## Productivity-mediated effects of composition

There is ample evidence that resident community productivity reduces the potential for the growth of invading species in microbes and non-microbes alike (*Crawley and Heard, 1999*; *Hodgson et al., 2002*; *Jousset et al., 2011*; *Kinnunen et al., 2016*). Cell density and respiration were likely important predictors of invasion resistance in our system because they were good proxies for how much the resident community had used the available resources that could otherwise have been used by the invader i.e. resource limitation. Interestingly, productivity at 7 days before (rather than immediately prior to) invasion had the most explanatory power in the models – suggesting that resource limitation may have operated in complex ways. One possibility is that those communities that could degrade the resource base most rapidly could also do so most efficiently, leaving less resource for the invader at 14 days. Alternatively, communities that have been at carrying capacity for longer periods may exhibit behaviours that select against invasion. For example, limited resources and/or metabolic stress can select for antibiotic production (*Craney et al., 2013*) or the production of inhibitory secondary metabolites (*Watrous et al., 2013*).

Structural equation modelling revealed that productivity strongly mediates the effect of starting composition on invasion success - further emphasising the extent to which invasion resistance is mainly a side-effect of productivity. This result is similar to that of previous studies with much more artificial microbial microcosms. *Hodgson et al., 2002* demonstrated a strong negative correlation

between *P. fluorescens* SBW25 invasion success and resident *P. fluorescens* SBW25 community diversity ($R^2$ = 0.7) in 2–5 strain communities grown in a defined laboratory medium, driven by the presence of the most productive *P. fluorescens* SBW25 strain type. Another more recent study with five-strain communities of *Ralstonia* grown in defined laboratory media, showed that resistance to invasion by another *Ralstonia* strain was determined by the presence of one or two strains with the fastest growth rate (*Yang et al., 2018*). What is remarkable about our results is not that we found a similar result to these studies per se, but that we found a similar result in microcosms containing a complex growth medium (beech leaf tea) inoculated with communities with richnesses of 66–236 OTUs. In our system as in more artificial microcosm experiments, it appeared that the starting abundance of particular group of bacteria – in our case, a putative 'functional group' containing several *Enterobacterales* (Functional Group 18) – determined the level of invasion resistance realised by the community (i.e. a dominance/selection effect).

The most parsimonious explanation for this effect in our system was simply that this was the most productive functional group, and a higher starting abundance gave this group of bacteria a better chance of outcompeting other bacteria and raising the level of growth in the community (i.e. a priority effect). Alternatively, the starting abundance of particular functional groups in a microcosm may determine community productivity in more complex ways – for example, because the most abundant early colonisers cause local environmental changes that affect the growth potential of other colonising bacteria (e.g. through cross-feeding and competition). In dental biofilms it has been demonstrated that a full biofilm can only be achieved with a certain order of colonisation, with metabolically-similar groups of early colonisers causing local environmental changes that allow for subsequent bacteria to colonise, allowing the full development of the biofilm (*Mazumdar et al., 2013*). Similarly, in a study system more similar to our own, it has been shown that the ability bacterial species to colonise a new phyllosphere environment depends on the local density (and by implication, identity) of other neighbouring bacteria colonising the environment at the same time – likely because more intense competition for resources reduces the average growth success of the population (*Remus-Emsermann et al., 2012*). Such species interactions/succession-type dynamics are likely to be at least partly driving productivity-mediated effects of composition in our system also – especially as our previous work has shown our 'functional groups' correspond to distinct metabolic profiles inferred from predicted metagenomes (*Pascual-García and Bell, 2020a*).

## Productivity-independent effects of composition

There was also some evidence that community composition impacted invasion resistance independently of the impacts of community productivity, although in a more minor way – as has been seen in a more limited sense in previous studies with artificial communities (*De Roy et al., 2013*; *Hodgson et al., 2002*).

One explanation for this is that groups of bacteria which were abundant in some communities but not others were functionally distinct; specialising on resource-poor or resource-rich environments (*Pascual-García and Bell, 2020a*) and degrading harder-to-access and/or rarer components of the resource pool. Previous work using the same study system has also shown that the starting composition of communities affects the extent to which recalcitrant substances are able to be used during subsequent community growth (*Rivett et al., 2016*).

Another possible explanation is that particular resident species acted as ecosystem engineers (*Pascual-García et al., 2020*). We have already mentioned the possibility of bacteria-driven environmental changes setting limits on the community productivity and leaving certain communities more vulnerable to invasion. Additionally, there may be other ways in which particular species engineer the environment to affect invader survival more directly - such as by modifying the pH of the environment or producing reactive oxygen species. Regarding pH modification, bacterial acidification has been shown to be an important factor in recent microbial invasion experiments in soil microcosms, which showed that *Pseudomonas* may be prevented from invading by the presence of species that alter the growth medium towards more acidic pH values (*Amor et al., 2020*). Regarding the production of reactive oxygen species, this has recently been demonstrated to be important in in-host studies of invasion resistance; *E. faecalis* has been shown to protect *C. elegans* nematodes from *Staphylococcus aureus* invasion by the producing reactive oxygen species which act as an antimicrobial to kill the invaders (*Ford and King, 2021*). However, whilst these more specialised mechanisms

of invasion resistance are likely to be fairly common, we emphasise again that they are likely to play a more minor role c more general factors such as community growth rate.

## Sub-hypotheses with weaker or no support

We found some evidence for the expected negative relationship between the diversity (Simpson's diversity of OTUs) of the inoculated community and the invaders' survival (*Figure 3*, *Figure 3—figure supplement 1*) in this semi-natural system, although this was relatively weak ($R^2$ = 0.09 and 0.06 for *P. fluorescens* and *P. putida*, respectively). Although it is hard to make a direct comparison, superficially at least, this weak relationship contrasted with previous experiments using artificial communities constructed from isolates or created by dilution-to-extinction of natural communities, where the relationship was clearer (*De Roy et al., 2013*; *Eisenhauer et al., 2013*; *van Elsas et al., 2012*). Nonetheless, given that previous experiments often found that the negative diversity-invasion relationship plateaued at higher diversities (*Bonanomi et al., 2014*; *Eisenhauer et al., 2013*; *van Elsas et al., 2012*), the weak negative slope observed in our high-diversity experiment may simply be because our communities all had relatively high levels of diversity. This suggests that species-like diversity is likely to be a less important predictor of invasion resistance in natural rather than artificial communities.

Phylogenetic diversity metrics had even poorer explanatory power, though this result should be interpreted with caution as gene trees based solely on the 16S gene have substantial limitations and estimating the phylogenetic diversity of natural communities is still a challenge (*Rajendhran and Gunasekaran, 2011*). Regardless, some preliminary conclusions about the importance of 16S phylogenetic diversity can be made. First, a phylogenetic equivalent of Simpson's diversity (Rao's quadratic entropy) was not related to invader survival, suggesting 16S phylogenetic alpha diversity is not predictive of invader survival. Secondly, the 16S-based phylogenetic distance of the communities from the invader (estimated number of mutations to between per nucleotide on their 16S gene) and the invaders' success only had a very weak positive relationship with invader survival (*Figure 2*). Darwin's 'naturalisation hypothesis' posits that invaders are more likely to establish when invading communities composed of resident species that are, on average, more phylogenetically distant from them, because of greater supposed niche differences (*Darwin, 1859*). However, evidence from previous bacterial microcosm experiments with artificial communities to support this hypothesis is mixed (*Gu et al., 2019*; *Jiang et al., 2010*; *Kinnunen et al., 2018*; *Li et al., 2019*), and we did not find strong evidence to support this hypothesis in natural communities. Somewhat related to the phylogenetic hypothesis, by using two different invaders in separate experiments, we also explored whether resident communities were better at resisting an invader when they had a high abundance of conspecific species (as was the case for *P. putida*; see Materials and methods: Laboratory techniques). Qualitatively, these results corroborate the idea that invaders with conspecific residents (*P. putida*) are less successful – although the general mechanisms of invasion resistance appear similar in both *P. fluorescens* and *P. putida* experiment (distribution of the variable importance values in *Figure 2*, shape of relationships between invader survival and the explanatory variables in *Figure 3*). A more dedicated study with more invader treatments would be needed to study this sub-hypothesis properly.

More specific measures of community phenotype related to bacterial metabolism were not informative. Again, this suggested that invasion resistance is primarily the result of generic rather than specific mechanisms.

## Unexplained variation

Finally, although we successfully explained much of the variation in invasion using the measured variables, a large component of the variation remained unexplained across the six sub-experiments. Given the predominant role of composition and productivity, we expect that some of this unexplained variation is due to the limitations of our techniques in capturing these key components of the resident communities.

Regarding composition, firstly, 16S amplicon sequencing was likely inadequate at capturing the microbial composition of the resident community inocula, and metagenomic/shotgun sequencing might have better characterised the compositional and functional diversity of communities and correlated with invasion. Secondly, although we tried to ensure that only bacteria were present in our communities (through filtering, passage and anti-fungal treatment, visual and flow-cytometric

observation; see Materials and methods: Field techniques), it is possible that some non-bacterial microorganisms such as fungi or phage survived and could have affected invader survival. This is difficult to avoid and so future studies might improve explanatory power by keeping the inoculated communities even more intact, characterising compositional effects on invasion at multiple taxonomic and trophic levels simultaneously (again using metagenomic sequencing). Thirdly, additionally characterising community composition nearer to the point of invasion may have improved explanatory power and better contextualised the results. Our unpublished work with a subset of these communities suggests that despite our lab acclimation period (see Materials and methods: Field techniques), some more minor compositional changes still occur in these communities after inoculation (as should be expected), with communities converging towards a similar composition but maintaining a high level of diversity. Characterising these changes and accounting for them in models of future experiments might therefore help confirm, for example, whether differences in communities' invasion resistance are the result of the rate at which certain functional groups of bacteria take over the community during growth.

Regarding productivity, the strong predictive power of productivity at 7 days (i.e. 7 days prior to invasion) implies that the temporal dynamics of the community growth is an important component of invasion resistance. Increasing the temporal resolution of growth measurements may therefore have improved the predictive power, as would methods to better distinguish active vs dormant proportions of the population.

Aside from productivity and composition, measuring additional variables in the growth period including pH (*Amor et al., 2020*) and better inferring functional performance using metagenomics and/or metabolomics could have also helped identify the role of environmental modification in preventing invasions. We also do not exclude the possibility that the unexplained variation may result from the stochastic nature of invasions, since stochastic processes can partly govern invader survival in microbial communities (*Amor et al., 2020*; *Kinnunen et al., 2018*).

## Conclusion

Experiments with artificial communities have suggested that the composition of microbial communities mostly affects invasion resistance by affecting their productivity (*Bonanomi et al., 2014*; *Eisenhauer et al., 2013*; *Hodgson et al., 2002*; *van Elsas et al., 2012*; *Yang et al., 2018*). Our experiment lends support to the extension of this claim to natural communities; semi-natural microcosms of bacteria mostly achieve invasion resistance primarily (though not wholly) as the result of their productivity, which is the result of the identity and abundance of the bacteria with which they are inoculated.

A new generation of microbial invasion ecology experiments with more natural microbial communities (*Bell, 2019*) might therefore benefit from placing a greater emphasis on productivity where diversity has historically been the focus of experimental designs. Productivity is arguably likely to be more variable than diversity among natural communities, because natural communities are already saturated with species and the population density of a community is more easily affected by environmental disturbances (e.g. dilution by rainfall, environmental pollution). Several lines of evidence in experimental microbial ecology exploring the effect of disturbances on invasion already suggest that disturbances primarily affect invasion success by affecting the population size of the resident community, for example (e.g. *Lear et al., 2020*; *Mallon et al., 2015b*). Equally, work outside microbial invasion ecology with pathogens frequently suggests that these 'invaders' frequently gain entry into host microbiomes by reducing the density of resident competitors (e.g. *Brown et al., 2008*; *Wei et al., 2018*). Efforts to characterise resident community growth trajectories before invasion – and what drives differences and disturbances to them - should thus be continued and extended in order to better understand microbial invasion resistance. We believe that taking resident community productivity as a null hypothesis for invasion resistance in this way will reveal more clearly the way invasion resistance emerges and is disrupted in natural microbial communities of all types (*Amalfitano et al., 2015*; *Fernandez-Gonzalez et al., 2021*; *Kinnunen et al., 2016*; *Litchman, 2010*; *Mallon et al., 2015a*; *Thakur et al., 2019*).

# Materials and methods

## Key resources table

| Reagent type (species) or resource | Designation | Source or reference | Identifiers | Additional information |
|---|---|---|---|---|
| Other | Communities | *Rivett and Bell, 2018*; 10.1038/s41564-018-0180-0 | NA | Cryopreserved tree hole communities archived in the lab of Professor Thomas Bell |
| Commercial assay, kit | ZR-96 DNA Soil extraction kits | Zymo Research Ltd | 11–324H | DNA extraction kit |
| Other | BLT | *Rivett and Bell, 2018*; 10.1038/s41564-018-0180-0 | NA | Bespoke culture medium made from Autumn/Fall beech leaves and water |
| Other | BD Accuri C6 Flow Cytometer | BD Biosciences | NA | Flow cytometer used for cell counts with Thiazole Orange (now discontinued) |
| Commercial assay, kit | MicroResp | The James Hutton Institute | 001 | Used for respiration assays |
| Commercial assay, kit | BacTiter-Glo | Promega | G8231 | Used for ATP assays |
| Chemical compound, drug | Xylose (β-xylosidase substrate); chitin (β-N-acetylhexosaminidase substrate); cellulose (β-glucosidase substrate); phosphate groups (phosphatase substrate) | Sigma-Aldrich | M7008; M2133; M3633; M8883 | Fluorescent substrates used for enzyme assays |
| Strain, strain background (*Pseudomonas fluorescens*) | SBW25 | Labs of Professors Thomas Bell and Craig MacClean; *Vogwill et al., 2016* | NA | Lux-transformed P. fluorescens SBW25 invader strain with IPTG-inducable luciferase reported gene |
| Strain, strain background (*Pseudomonas putida*) | KT2440 | Labs of Professors Thomas Bell and Craig MacClean; *Vogwill et al., 2016* | NA | Lux-transformed P. putida KT2440 invader strain with IPTG-inducable luciferase reported gene |
| Chemical compound, drug | IPTG | Sigma-Aldrich | I6758 | Needed to induce luminescence in the lux-tagged strains |
| Software, algorithm | R; RStudio | R Project for Statistical Computing; RStudio | RRID:SCR_001905; RRID:SCR_000432 | Used for the majority of data wrangling and analysis. |
| Software, algorithm | Geneious 2.0 | Biomatters Ltd | RRID:SCR_010519 | Used for construction of phylogenetic tree. |
| Software, algorithm | Functional group abundances | APG's Github repository version 1.0.0 deposited in Zenodo [DOI: 10.5281/zenodo.5562687; (*Pascual-García, 2021*) this paper's OSF repository https://doi.org/10.17605/OSF.IO/HC57W] | NA | Functional group abundances and the computational methods used to produce them (APG's Github repository). |

## Field techniques

### Field sampling of communities

We used naturally occurring bacterial communities collected from 680 tree hole communities across Southern England between August 2013 and April 2014. The term 'tree hole' refers to the naturally occurring, semi-permanent pools of rainwater that collect in the pans formed by the buttress roots of *Fagus sylvatica* beech trees (*Bell et al., 2005a*). Bacterial communities inhabit these phytotelmata, thriving on the organic matter (the bulk of which is beech leaf litter) that collects in them. Communities were sampled in the field by actively searching for pools of water in the buttress roots of beech trees. Once located, GPS coordinates and date of collection was recorded, the tree holes were

homogenised using a sterile plastic Pasteur pipette, and 1 ml was transferred into a sterile 1.5 ml centrifuge tube (Starlab; Milton Keynes, UK). Samples were transported to the lab at an ambient temperature within 24 hr of collection, where they were diluted 1:4 in sterile phosphate buffered saline and filtered (pore size 20–22 μm, Whatman four filter paper) to remove debris and large organisms. 500 μl of this filtrate was then used to inoculate 5 mL of beech leaf tea medium (see 'Laboratory culturing of communities' for a description) containing 200 μg ml$^{-1}$ cyclohexamide (Sigma-Aldrich) to remove fungi, and the communities allowed to grow for 7 days at 22°C to reach stationary phase and acclimatise to laboratory conditions. Communities were then frozen in 60% glycerol with NaCl solution (Sigma-Aldrich; Gillingham, UK) to produce a frozen community archive, and revived from these frozen stocks for further experiments.

Thus, these communities were 'natural communities' to the extent that they had only been subject to manipulations for practical purposes – dilution for filtering, the removal of fungi, acclimation to the laboratory growth medium, and freezing/thawing. We consider that the 1:4 dilution for filtering was minor compared to that typically used in dilution-to-extinction experiments (typically 90% at each dilution step). The removal of fungi as a practical simplification of the communities is perhaps most contentious manipulation, as fungi are likely to be important decomposers of leaf litter in this system. However, previous experiments also typically worked with bacteria-only resident communities and so our results are comparable. Finally, freezing the communities and re-growing them from frozen stocks for each experiment was a practical step to allow repeatable experiments using the same, fully sequenced starting community. To ameliorate this, the whole procedure of re-growing, functionally characterising and invading communities was repeated in four separate assays and the means of these four pseudo-replicated measurements used for the final analysis.

## Laboratory techniques
### Amplicon sequencing of inocula communities
Communities were sequenced at the point of freezing/archiving and thus, when we talk about 'composition' in this paper, we are referring to the composition of the inocula added to the microcosms at the start of the experiment – as is conventional for microbial diversity/composition-invasion resistance experiments. Briefly, immediately prior to archiving, DNA was extracted from all communities using ZR-96 DNA Soil extraction kits (Zymo Research Ltd, Irvine, CA, USA). Extractions were amplified on the V4 region of the 16S rRNA gene, using the barcoded 515F and 806R PCR primers using the HotStarTaq Plus Master Mix Kit (Qiagen; Valencia, CA, USA). The PCR cycle used was: 94°C for 3 min, followed by 28 cycles of 94°C for 30 s, 53°C for 40 s and 72°C for 1 min, followed by a final elongation step at 72°C for 5 min. Sequencing was performed by MR DNA using the MiSeq platform (https://www.mrdnalab.com; Shallowater, TX, USA), with primers and barcodes removed according to their standard procedure.

### Laboratory culturing of communities
The common garden culture medium selected for this experiment was a 'beech leaf tea' (BLT), designed to approximate the predominant nutrient conditions found in tree holes. To produce this medium, 50 g of autumn-fall beech leaves were autoclaved with 500 ml of distilled water to create a concentrated solution that was diluted 32-fold to produce the final BLT culture medium. To begin the common garden experiment, communities were revived from the acclimatised stocks by adding 50 μl of each stock to 1.8 ml of BLT in 1.8 ml deep-well 96 well plates and growing for 14 days at 22°C before invasion treatments were applied.

### Productivity measures
During the 14 day growth period prior to invasion, cell density and respiration were characterised at 7 (7 days before invasion) and 14 days (the day of invasion) as estimates of productivity.

Cumulative density of the communities was measured by sub-sampling the communities and staining with Thiazole Orange (Sigma-Aldrich; Gillingham, UK) at concentration of 100 nM for 15 min and analysing cultures using flow cytometry (BD Accuri C6; BD Biosciences, San Jose, CA, USA). Thiazole Orange is a membrane permeant nucleic acid stain, staining both live and dead bacteria and thus this assay represents the total cumulative yield in each community. Fluorescence gating against

negative (beech tea) controls was used to count the number of fluorescent cells in each community (cumulative cell density).

Respiration of each community was measured by sub-sampling and using the MicroResp system (The James Hutton Institute; Aberdeen, UK). Briefly, agar-set indicator gels were suspended above growing sub-cultures at 7 and 14 days in a deep-well 96-well plate in an airtight system. As growing cells respired, $CO_2$ released was absorbed by the indicator gels above the cultures, and the gel colour changed from pink to purple. Colour change is read via a spectrophotometer prior to suspension above cultures (OD600 at 0 hr), and after 24 hr of respiration/after suspension above cultures. The change in colour is used to calculate $mgCO_2$ released in the 24 hr period using a standard curve.

## Additional phenotypic measurements

We also took phenotypic measurements related to the metabolic activity of communities at 7 and 14 days by performing assays of ATP and enzyme activity. ATP activity assays provide a general estimate of how metabolically active communities are, and we expected that communities that were more metabolically active (i.e. producing more ATP) would be more able to actively defend against invaders (e.g. through resource competition or direct competition).

ATP activity assays were performed at 7 and 14 days of community growth prior to invasion using the BacTiter-Glo Cell Viability Assay (Promega; Madison, WI, USA). This assay is a two-step process that releases ATP stored inside cells in order that it can bind to bind to ATP-activated luciferase present in the formula. The maximum luminescence generated in the immediate 5 min reading period is thus proportional to the amount of ATP in the sample, which is calculated exactly by converting into concentration in nM using a standard curve.

Enzyme activity assays measured the degree to which communities were degrading particular substrates present in the BLT. We measured the metabolism of the following substrates by the following enzymes; hemi-cellulose by β-xylosidase, chitin by β-N-acetylhexosaminidase, cellulose by β-glucosidase and phosphate groups by phosphatase. These substrates are common in beech leaf litter (*Aneja et al., 2006*), the complex substrate constituting the BLT medium. Substrates labelled with the fluorescent moiety 4-methylumbelliferone (MUB) (Sigma-Aldrich; Gillingham, UK) were incubated with communities at a working concentration of 400 M for 1 hr, as per *Frossard et al., 2013*. Fluorescence is generated when labelled substrates are cleaved and deprotonated by bacterially produced enzymes. Fluorescence detected spectrophotometrically is thus used to calculate mg/ml of each enzyme associated with each substrate present in each community – a measure of the capacity of communities to metabolise certain substrates in the BLT.

## Invasion assays

After the 14 day growth period during which productivity and metabolic activity were characterised, two lux-tagged, IPTG-inducable bacteria - *Pseudomonas fluorescens* SBW25 and *Pseudomonas putida* KT2440 – were separately introduced into each of the 680 communities. To perform these two invasion assays, communities were homogenised and aliquoted out twice into 237.5 μl volumes in sterile white microtitre plates. Each invader was grown in 10 ml of BLT medium for 96 hr prior to invasion, and 10 μl added to the community aliquots with 2.5 μl of 100 mM IPTG solution (final concentration 1 mM IPTG). Invaders reached mean densities of $2.5 \times 10^7 \pm$ cells/mL in 48 hr, and so this represented a high invasion pressure of approximately $10^5$ invader cells entering each community containing an average of $10^5$ cells according to cytometry readings.

## Choice of invaders

*Pseudomonas* were chosen as invaders both because of the aforementioned evidence of their abundance on beech leaves, and because they are fast-growing, metabolically versatile bacteria that are common colonisers, contaminants and pathogens in microbial communities (*Nikel et al., 2014*). Furthermore, *Pseudomonas* is one of the most cosmopolitan bacterial genera, its members being found in many very diverse environments (*Pascual-García et al., 2014*) and being a significant component of the bacterial flora of leaves (e.g. *Aneja et al., 2006*), especially just prior to the beginning of litterfall where they represent approximately $10^6$ cells per gram according to one available estimate (*Holm and Jensen, 1972*). Combined with the high invasion pressure, we therefore considered them

good candidates to be successful invaders, allowing us to study invasion resistance under conditions where it was likely to be most possible and detectable, and invader-side factors were less limiting.

The choice of two closely related species was initially motivated by the intention to explore, in a minor sense, whether invasion resistance and its mechanisms were different between communities that did not have a high abundance of conspecifics to the invader and those that did. None of the OTUs present in the resident communities was aligned to *Pseudomonas fluorescens* whilst one was aligned to *Pseudomonas putida* and had a mean abundance of 2.35% ± 0.02 (and so *P. fluorescens* represented a more 'true' invader). As detailed in our Results, anecdotally our results support the idea that invasion resistance is greater where there is a high abundance of conspecifics, but that the mechanisms of invasion resistance are similar regardless. However, there could have been many more reasons for this and species-level resolution for OTUs is unreliable (*Janda and Abbott, 2007*; *Johnson et al., 2019*; *Knight et al., 2018*). In order to support this hypothesis robustly a more dedicated study would need to be done, probably supported by metagenomic sequencing approaches.

### Invader survival

Luminescence of lux-tagged bacterial strains is directly related to the density of metabolically active cells (*Close et al., 2012*). Therefore, luminescence values of each microcosm were used to calculate the number of metabolically active invader cells (invader survival) produced from the original invader inoculum. As described previously (*Jones et al., 2017*), we calibrated the luminescence values by performing growth assays of the invader in BLT medium with IPTG (1 mM) prior to the experiment, measuring luminescence and plating cultures onto LB agar to obtain cell numbers. There was a strong relationship between $\log_{10}$ luminescence and $\log_{10}$ plate counts (R2 = 0.87 and 0.82 for *P. fluorescens* and *P. putida*, respectively) during log phase, so we used the calibration curve to convert luminescence values into invader cell densities. Invader survival was defined as the density of metabolically active invader cells/ml$^{-1}$ in the resident communities after some defined period since invasion (24, 96, and 168 hr). The upper detection limit of the luminescence assay was 12 lumens (the maximum background luminescence of 1000 sterile BLT microcosms), which equates to estimated invader survival detection limits of $6.26 \times 10^3$ cells/mL for *P. fluorescens* and $1.06 \times 10^4$ cells/mL for *P. putida*. These detection limits were accounted for in our sensitivity analysis (see Materials and methods: Statistical techniques) which suggested that values below the detection limit contained some real invader signal as well as noise.

In total, therefore, we assayed 680 communities (n = 680) for invasion resistance to two invaders with invader survival measured at three time points (680 x 2 x 3 = 4063 measurements total). 680 was the number of communities remaining from the original set of 753 communities after the removal of communities with less than 10,000 sequences and without all growth assay and/or invasion assay data. This entire 3 week long assay of 4063 measurements (2 weeks of community growth, followed by invasion and measurement of invader survival up until 1 week after invasion) was repeated four times (technical replicates = 4) from frozen stocks to the last measurement (invader survival at 168 hr after invasion). Values used in the experimental analysis are the means calculated from these technical replicates. No explicit power analysis was used as our study was not straightforwardly hypothesis-driven (or at least there were many possible hypotheses) and we had a very large number of replicates (n = 680).

## Computational techniques

### Diversity metrics calculated from compositional data

Sequenced communities were characterised with bioinformatics implemented at the sequencing centre (Molecular Research DNA; https://www.mrdnalab.com) as described previously (*Rivett and Bell, 2018*). Briefly, sequences under 150 bp and/or with ambiguous calls were removed, and then the remaining sequences were subjected to de-noising and chimera removal. Sequences were then classified into operational taxonomic units (OTUs) at 97% similarity. Subsequently, extremely rare OTUs occurring in less than 10 samples or with under 100 reads across all samples were removed. We also removed communities with less than 10,000 reads/sequences per sample. This process produced an OTU abundance table of 581 OTUs with the abundance of OTUs being the number of sequences per OTU (*Source data 1*, *Figure 2—figure supplement 1*). We used this OTU abundance table for all downstream analysis relating to community composition. OTU diversity (Simpon's Index)

was computed from the species abundance table using the diversity function from the R package vegan (*Oksanen et al., 2019*). Simpson's Index was chosen over Shannon's because it gives more weight to relative abundance vs. species richness, and many OTUs had single digit abundances in some communities. Inspection of exploratory plots of invasion against the two diversity indices also revealed that there was a clearer correlation between invader survival and Simpson's diversity.

The two phylogenetic metrics were calculated based on the OTU abundance table and a phylogenetic tree we constructed using the 16S sequence associated with each of the 581 OTUs, as well as the two sequences associated with each of the two invaders and the archaea *Halobacterium salinarum* (obtained from NCBI). These sequences were then trimmed to 250 bp and aligned using MUSCLE multiple sequence alignment in Geneious 2.0. A phylogenetic tree was created with a Maximum Likelihood method of phylogenetic inference using RaxML with 100 bootstraps using the GTR+G sequence evolution model. After construction, this tree was rooted by specifying *Halobacterium salinarum* as the outgroup. It is important to briefly note the limitations of using such a 16S gene tree when interpreting our results about phylogenetic effects. Whilst 16S is arguably the optimum single-locus choice for microbial phylogeny analysis, 16S tree trees' efficacy is limited by problems such as horizontal gene transfer, different versions of 16S genes within a single organism and the large differences in the similarity of 16S genes among species depending on the genus considered (*Rajendhran and Gunasekaran, 2011*). Our phylogenetic tree had broad consensus with the assigned genus-level taxonomy of the OTUs, but nonetheless there were inconsistencies and the presence of six extremely long branch lengths (>99th percentile) suggested that a better phylogeny is needed to make firmer conclusions about the importance of phylogenetic composition for invasion resistance.

Using this 16S phylogenetic gene tree, a phylogenetic diversity metric (Rao's quadratic entropy) was computed from the species abundance table using the diversity function from the function rao. diversity from the R package SYNCSA (*Debastiani and Pillar, 2012*). Rao's quadratic entropy is a phylogenetic equivalent to Simpson's diversity, giving the mean phylogenetic distance between any two randomly chosen individuals (in this case OTUs) in the community, taking into account their relative abundance (*Rao, 1982*). Phylogenetic distance between the invader and community was calculated as follows. First, the cophenetic function of the picante package (*Kembel et al., 2010*) was used to calculate the pairwise distances between all OTUs in the master phylogenetic tree and each of the invaders, resulting in two vectors of the phylogenetic distance (number of branch lengths; number of inferred substitutions per 16S gene) between each invader and all OTUs. Second, these vectors were used to calculate an abundance weighted invader-community phylogenetic distance metric using the weighted.mean function (stats package; *R Development Core Team, 2020*) to calculate the phylogenetic distance between the invader and each OTU, weighted by the relative abundance of that OTU in each community.

## Dimensionality reduction approaches tested

Given the difficulty of analysing the effects of 581 OTUs with only 680 samples, we opted to reduce the dimensionality of the starting composition data before proceeding with our main analyses, testing seven approaches.

For the first five approaches, we simply aggregated the abundances of the OTUs at each of five taxonomic ranks (360 genera, 177 families, 90 orders, 42 classes, 18phyla), according to the species-level taxonomic assignment of the OTU. This approach was intended to account for the fact that higher level taxonomic groupings may drive compositional effects (e.g. abundance of a clade with a particular functional role) and relatedly, take account for the fact that species-level resolution for OTUs is unreliable (*Janda and Abbott, 2007*; *Johnson et al., 2019*; *Knight et al., 2018*).

For the sixth approach, we performed a principal coordinates analysis of the distance matrix of the Jensen-Shannon divergence (*Endres and Schindelin, 2003*) of OTU abundances using the 'dudi. pcoa' function from the ade4 package (*Chessel et al., 2004*; *Figure 2—figure supplement 2*), selecting the first five coordinates by inspecting the scree plot.

For the seventh and final approach, we calculated a network representing significant correlations between OTU abundances inferred with SparCC (*Friedman and Alm, 2012*), considering only significant links (pseudo p-value<0.01, absolute value of the correlation coefficient > 0.2) leading to a network whose links represent either segregations (negative correlations) or aggregations (positive

correlations) between species. Note that these links do not necessarily reflect ecological interactions, since correlations may be caused by bacteria's environmental preferences (*Pascual-García et al., 2014*). We next partitioned the network into functional groups using functionInk (*Pascual-García and Bell, 2020b*). Briefly, functionInk is a community-detection method that classifies the nodes in the network clustering together those nodes sharing approximately the same number and type of links (aggregations and segregations in our case) with respect to the same neighbours. Since the nodes have the same links with the same neighbours, we say they have the same functional role in the network, hence the name of functional groups. In previous work, we observed that a classification of the samples according to their beta-diversity distance, led to the identification of six community-classes, and that these classes had distinct functional and metagenomic signatures (*Pascual-García and Bell, 2020a*). To identify these functional groups (*Supplementary file 1*), we run functionInk on the network finding an optimal partition at the maximum of the total partition density (*Pascual-García and Bell, 2020b*) with 63 functional groups. For the final analysis, we selected groups that had three or more members, plus a group of functional groups of *Paenibacillus* species that had less than three members. This was done on the basis of the high abundance of *Paenibacillus borealis* across the communities, and the high connectivity of *Paenibacillus* OTUs. This whole process resulted in the selection of 19 functional groups (+ 1 variable representing the total mean abundance of remaining OTUs) as the representation of composition for this approach.

## Selection of a dimensionality reduction approach

To select the most appropriate dimensionality reduction approach for our analysis, we compared their explanatory power for invader survival using random forest regressions (see below). To do this, before our main analyses, we computed 7 sets of random forest regressions using only these dimensionality reductions of starting composition as the explanatory variables, and *P. fluorescens* and *P. putida* invader survival at 24, 96, and 168 hr (1, 3, and 7 days since invasion) as the response variable. In total, there were therefore 21 random forests compared (seven dimensionality reduction approaches x 2 invaders x three sampling points). We compared the dimensionality-reduction techniques by plotting the total variance explained against the number of variables permuted in each of the 21 random forests (*Figure 2—figure supplement 3*). This revealed functional groups approach (*Pascual-García and Bell, 2020b*) to be the most appropriate dimensionality reduction of composition to explain invasion, offering the best compromise in terms of variance explained vs. number of variables permuted. We therefore proceeded with using the abundances of the 19+1 functional groups as explanatory variables in the main analysis random forests, adding the phenotypic explanatory variables alongside them to explain invader survival.

## Statistical techniques

All statistical analyses were performed in the R programming language (*R Development Core Team, 2017*). Before carrying out analyses, we visually inspected plots of the two-way relationships between invader survival and each of the explanatory variables under various transformations to identify appropriate transformations. The response variable, invader survival, was $\log_{10}$ transformed after adding a pseudocount (+1) to each of the density/ml estimates, in order to deal with zero values. All the explanatory variables except for the three diversity indices (Simpson's diversity and phylogenetic distance from the invader) were $\log_{10}$ transformed.

## Random forest regressions

We used random forest regressions implemented using the 'randomForest' package (*Liaw and Wiener, 2002*) to find the most important individual explanatory variables for invader survival among many. The main advantage of the random forest approach for this was that it is able to deal well with overfitting associated with having many explanatory variables, by randomly shuffling the explanatory variables (and samples) permuted in each constituent regression tree and calculating their mean importance across the 'forest'. It also makes few assumptions about the relationships between dependent and independent variables than parametric approaches, accounting somewhat for non-independence and non-linearity, for example.

For the main analysis, random forests were computed for each of the six sub-experiments (2 invaders x three time points). For each community, starting composition was computed using the

selected dimensionality reduction approach i.e. the mean sequence abundance (across species in the group) in each of the 19 (+1) functional groups). Additionally, the 14 phenotypic measurements of the communities were included as explanatory variables, totalling 37 explanatory variables. We used the variable importance values from the random forests to narrow down the 37 explanatory variables to the best predictors of *P. fluorescens* and *P. putida* invader survival.

## Structural equation models

The random forest approach was useful for identifying the key explanatory variables for invader survival in our system, but as a black box approach, it is difficult to use to understand the shape of the relationships between explanatory and response variables, and assess fit. In our second part of our main analysis therefore, we sought to estimate the putative chain of causation between composition, productivity and invasion resistance using structural equation modelling (SEMs). This approach allowed us to test three scenarios to estimate the extent by which the productivity mediated the effect of composition on invasion (see main text). SEMs were implemented using the lavaan package (*Rosseel, 2012*).

For these models, we created a latent variable called 'Productivity' including cell yield at 7 and 14 days. We chose these variables because in two out of the six of experiments (those with the highest average invader survival) cell yield and 7 days was most important, whilst in the other 4 (those with the lowest average invader survival) cell yield at 14 days (the day of invasion) was most important (*Figure 2*). Although important in the random forests, respiration was not included in the SEMs because we considered that respiration must result from cell density/growth rate, and cell yield measurements were always the most important explanatory variables in random forests. We created another latent variable called 'Composition', including the abundance of the 19 (+1) functional groups and the Simpson's diversity of the OTUs. We included all 19 (+1) functional group abundances and diversity because although Functional Group 18 was clearly the most important determinant of invader survival, the importance values for starting composition variables were more evenly distributed than those for the phenotypic measurements, where productivity measures were clearly the most important (*Figure 2*). Finally, we created an 'Invasion' latent variable encapsulating all measurements of invader survival for both invaders at the three sampling points (24 hr, 96 hr, and 7 days after invasion).

The chosen explanatory variables (the abundance of the 19+1 functional groups, Simpson's diversity, and 2 measures of resident community cell density at 7 and 14 days) were entered as latent variables in these models to explain the six invader survival end-points (response variables), after the same $\log_{10}$-transformations applied previously (ensuring the variances were approximately equal/within two orders of magnitude). The three structural equation models described in Results (full structures presented in *Figure 4—figure supplements 1–3*) were then built and analysed with the R package 'lavaan' (*Rosseel, 2012*). Although the fit of the three models could be improved specifying residual correlations guided by the computation of modification indexes (*Lin et al., 2017*), we omitted this step. This optimisation should be done for each model independently and it is unclear how the process could be performed in parallel in a way in which a fair comparison between the final (optimised) models is guaranteed. Hence, we compared the three basic models in terms of their AIC values, and an ANOVA test was performed to evaluate if the reduction in degrees of freedom of the partial mediation model with respect to the complete mediation model implied a significant reduction in the AIC value (*Barrett, 2007*). We also verified that other metrics such as the Comparative Fit Index were also improved (*Bentler, 1990*). SEMs were visualised using base functions (*R Development Core Team, 2020*), the package semPlot (*Epskamp, 2015*) and the Arrows function from 'shape' (*Soetaert, 2020*).

We conducted sensitivity analyses as part of the SEM analysis, in order to validate that the results of the main analysis on the raw data were not an artefact of random noise below the detection threshold of the luminescence assay (which was converted into CFU/ml units). For this analysis, we randomly shuffled the under detection limit invader survival values 999 times by sampling the CFU-converted under detection limit data without replacement using the 'sample' function (base package). To assess sensitivity, we examined the average model fit parameters for the Complete, Partial and No Mediation models fit to the 999 generated datasets. When values below the detection limit were randomly shuffled, the model comparison very marginally supported the Complete (mean

ΔAIC = 0.97) over the Partial mediation model (mean ΔAIC = 0.02). However, the AIC values of all three models also increased by 1383.71–1400 (i.e. fit decreased) - indicating that at least some the values below the conservative detection limit were informative in the original models. Despite the sensitivity of the result therefore, the nature of the change was consistent with the idea that productivity-independent effects of composition played a role in invasion resistance alongside effects of productivity.

## Other R packages used

Other R packages used include 'rstudioapi' for setting working directory (*Ushey et al., 2020*), 'tibble', 'stringr' and 'plyr' for data wrangling (*Müller and Wickham, 2020*; *Wickham, 2019*; *Wickham, 2011*), 'plotrix' for standard error calculation (*Lemon, 2006*), 'caret for extracting variable importance values from random forests (*Kuhn, 2020*), RColorBrewer for figure production (*Neuwirth, 2014*), ape for phylogenetic tree wrangling (*Paradis and Schliep, 2019*) and 'grateful' for generating a bibliography of the packages used (*Rodríguez-Sánchez and Hutchins, 2020*).

## Data availability

Data and R code is available via OSF and Github, copy archived at https://doi.org/10.5281/zenodo.7640512 (*Jones et al., 2021*).

## Acknowledgements

The funders of this research were the European Research Council (ERC StG 311399-Redundancy), and APG was also funded by the Simons Collaboration: Principles of Microbial Ecosystems (PriME, award number 542381). Thank you to the funders and the EU more generally for making this PhD and research possible. Thank you to members of the Bell, Barraclough and Raymond Labs for support throughout this project. Thank you to Professors Tim Barraclough and Jonathan Jeschke, who provided useful comments to improve this work as PhD viva examiners. Finally, thank you to the manuscript's reviewers, who provided very helpful feedback to significantly improve this manuscript.

## Additional information

### Funding

| Funder | Grant reference number | Author |
|---|---|---|
| H2020 European Research Council | ERC StG 311399-Redundancy | Matt Lloyd Jones<br>Damian William Rivett<br>Alberto Pascual-García<br>Thomas Bell |
| Simons Foundation | 542381 | Alberto Pascual-García |

The funders had no role in study design, data collection and interpretation, or the decision to submit the work for publication.

### Author contributions

Matt Lloyd Jones, Conceptualization, Data curation, Software, Formal analysis, Validation, Investigation, Visualization, Methodology, Writing - original draft, Project administration, Writing - review and editing; Damian William Rivett, Conceptualization, Resources, Data curation, Supervision, Investigation, Methodology, Project administration, Writing - review and editing; Alberto Pascual-García, Resources, Data curation, Software, Formal analysis, Supervision, Methodology, Writing - review and editing; Thomas Bell, Conceptualization, Resources, Supervision, Funding acquisition, Methodology, Project administration, Writing - review and editing

### Author ORCIDs

Matt Lloyd Jones (iD) https://orcid.org/0000-0001-5841-4554
Damian William Rivett (iD) https://orcid.org/0000-0002-1852-6137
Alberto Pascual-García (iD) https://orcid.org/0000-0002-8444-3196

**Decision letter and Author response**
Decision letter https://doi.org/10.7554/eLife.71811.sa1
Author response https://doi.org/10.7554/eLife.71811.sa2

# Additional files

### Supplementary files

• Supplementary file 1. Functional group membership – taxonomic assignments for each of the OTUs in each of the 18 functional groups identified using the functionInk approach.

• Source data 1. OTU table for the 680 communities used in the analysis. OTU names are their species-level taxonomic assignments.

• Transparent reporting form

### Data availability

All composition, phenotypic and invasion assay data (raw inputs and processed outputs) are deposited at Open Science Framework (https://doi.org/10.17605/OSF.IO/HC57W). These data can be used alongside the R code deposited at the lead author's GitHub repository for this study (https://github.com/befriendabacterium/communityinvasion) to reproduce the analysis described in this paper. These data and code are archived together at Zenodo (https://doi.org/10.5281/zenodo.7640512). The raw sequence data underpinning the compositional data was previously deposited in the NCBI Short Read Archive with accession number SRP145037.

The following dataset was generated:

| Author(s) | Year | Dataset title | Dataset URL | Database and Identifier |
|---|---|---|---|---|
| Jones ML, Rivett DW, Pascual-García A, Bell T | 2021 | Relationships between community composition, productivity and invasion resistance in semi-natural bacterial microcosms | https://doi.org/10.17605/OSF.IO/HC57W | Open Science Framework, 10.17605/OSF.IO/HC57W |

The following previously published dataset was used:

| Author(s) | Year | Dataset title | Dataset URL | Database and Identifier |
|---|---|---|---|---|
| Rivett DW, Bell T | 2018 | Abundance determines the functional role of bacterial phylotypes in complex communities | https://www.ncbi.nlm.nih.gov/sra/?term=%20SRP145037 | NCBI Sequence Read Archive, NCBISRP145037 |

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
