## [Decision Letter]

**Acceptance summary:**

Jones and colleagues have experimentally tested whether residential community composition, species richness or productivity of several hundred bacterial communities determine the invasion success of two strains of *Pseudomonas*. Productivity was a key predictor of invasion resistance, mediating the effect of composition on invasion resistance. The particular relevance of this study is that the authors used natural bacterial communities isolated from tree holes.

**Decision letter after peer review:**

[Editors’ note: the authors submitted for reconsideration following the decision after peer review. What follows is the decision letter after the first round of review.]

Thank you for submitting your work entitled "Productive bacterial communities exclude invaders" for consideration by *eLife*. Your article has been reviewed by 3 peer reviewers, one of whom is a member of our Board of Reviewing Editors, and the evaluation has been overseen a Senior Editor. The following individual involved in review of your submission has agreed to reveal their identity: Sofia van Moorsel (Reviewer #2).

Our decision has been reached after consultation between the reviewers. Based on these discussions and the individual reviews below, we regret to inform you that your work will not be considered further for publication in *eLife*.

As mentioned in the review of the Reviewing Editor, we believe that your study would have great potential for an entirely new submission, which we would allow if you can resolve the issues mentioned by the reviewers. The novelty of your study is that you took natural communities into the lab and tested under uniform conditions how their diversity affected the time to exclusion of an internal vs. an external invader, and how this effect of community composition on ecosystem functioning was mediated by community productivity. However, your analysis and writing do not reflect this narrative but rather compare the effects of some minimal proxies of community composition/diversity with those of the covariate community productivity, excluding the link diversity-productivity. Furthermore, it is not clear if you bacterial communities are only containing bacteria or also fungi or other organisms. For further details please carefully study the three attached reviews.

*Reviewer #1:*

I have some mixed feelings about this manuscript. On the one hand, I like the fact that natural microbial communities, rather than experimentally assembled communities or communities established through serial dilution, were used to study microbial invasions. Experimentally assembled microbial communities are much simpler than natural communities, and the use of serial dilution to generate diversity gradient runs the risk of confounding species diversity effects with species composition effects. The use of natural communities that differ in diversity and composition eliminates these problems. On the other hand, however, there are several issues in the manuscript that complicate the interpretation of the reported data.

First, if I am not mistaken, it seems that no treatment was done to eliminate fungi in the water samples collected from tree holes, thus fungal communities were likely present in the experimental microcosms. If this is the case, fungi may have competed with bacteria for resources, and played a role in regulating bacterial invasion in the experiment. Thus although this study did not mention fungi at all, the reported findings may be vulnerable to alternative explanations related to the role of resident fungal communities for bacterial invasion.

Second, the use of only two closely related *Pseudomonas* species as invaders invites question about the generality of the reported findings. This small number of invading species contrasts sharply with the large number of different resident communities used in the experiment. A greater number of bacterial species, with different degrees of evolutionary relatedness, should have been used to ensure the robustness of the findings.

Third, the choice of the large inoculum size (106 individuals per ml for each invader) is also puzzling. The initial population size of the invaders actually was so high that they were probably greater than population sizes of most, if not all, resident species. Such large propagule pressure almost never exists in nature, making this study an unrealistic exploration of biological invasions. Apparently, the two invader species were competitively inferior to some of the resident species, and declined over time in abundance after their introduction, and some invader populations were probably on their way to extinction. In this context, the term "invasion success", quantified as invader abundance, is also misleading.

Overall, I feel these issues are significant and cannot be addressed with revision, and recommend the rejection of this manuscript for publication in *eLife*.

*Reviewer #2:*

In their paper, Jones et al. conducted an experiment testing whether residential community composition, species richness or productivity of several hundred bacterial communities determines the invasion success of invading bacterial species (two strains of *Pseudomonas*). The study is innovative, because the authors used natural bacterial communities isolated from tree holes (which they have used in previous studies, too).

The found several community types that cluster according to species composition and two of types were particularly vulnerable to invasion, even though the species diversity was comparable to the other community types. This difference in community composition lead to differences in community productivity, which then in turn determined invasion success of the two invaders. The authors found that community diversity explained only negligible part of the invasion success. These findings have large implications for the field of microbial invasion ecology. The authors argue convincingly that future studies should go beyond the idea of community diversity and community composition by measuring as many community functions as possible and by conducting such experiment in a more dynamic system.

The figures are great. They are carefully prepared, easy to understand and they can be interpreted without reading the full paper.

1. Impact statement: When reading through the abstract and the introduction, it seems to be that the statement should rather read "A microbial community's productivity is a better predictor of its invasion resistance than its diversity".

To me it seems that productivity is better than composition, but composition is still much better than diversity. For example, see statement on lines 106/107. This result is a bit lost in the title and impact statement. Or maybe the introduction should be targeted a bit more towards composition.

2. L. 202: 48% is very high! Many researchers can only dream of such explanatory power.

3. L. 228: This is the only paragraph directly following up the Introduction, whereas the rest of the Discussion wasn't introduced as thoroughly. I definitely suggest rewriting the Introduction to move away from the pure diversity focus, bringing in composition, how composition and growth may be related, and maybe even the naturalisation hypothesis.

4. L. 327-33: This is written in an unnecessarily convoluted and complex fashion. I suggest rephrasing to get the main message better across.

*Reviewer #3:*

As a plant ecologist and biodiversity scientist myself, I cannot judge some of the technical issues and therefore assess this manuscript from my own point of view. In fact, I think the authors did a very novel and excellent study in the still and ever-increasing, hot biodiversity-ecosystem functioning topic. But they do not take sufficient advantage of it (1) and suffer from a fundamental misconception (2).

Regarding (1): experiments manipulating species richness have shown that reducing diversity in a working ecosystem generally reduces its functioning. In contrast, it is less clear if natural (i.e. working) ecosystems of higher species richness function better than natural ecosystems with lower richness. One of the main problems are confounding environmental variables such as soil fertility. Thus, the novel experiment that nobody had done so far would be to take the species of these natural ecosystems and recreate them under common environmental conditions, e.g. by planting plant communities in the same composition in an experimental garden as they occur in their natural environments. The study done by the authors is the first ever doing almost exactly this and thus super novel (I'm actually surprised myself that we never had the idea to do this novel experiment with plants!). I say "almost exactly", because it is not clear to me how well the authors could know that (a) their OTUs are species and (b) their communities contained only those species, i.e. only bacteria and no other organisms (see also comment by second reviewer).

Regarding (2): there have been long, often misguided discussions about

the importance of "species richness per se" in affecting ecosystem functioning vs. other factors, in particular species identities and species composition. The authors take this even further and compare the importance of diversity (including both richness and composition aspects, say D) with the importance of an "ecosystem function"(productivity proxies, say P) as covariate on invasion success, i.e. ecosystem functioning (say EF). This is like earlier arguments saying that when assessing effects of plant species richness (D) on community biomass (EF) one should consider only plots with high plant cover (P). However, the hypothesis is D → EF, and for that to be true it does not matter which mechanism (and how), in this case D → P → EF, mediates the causal relationship D → EF. When "importance" of explanatory variables is assessed, one cannot compare mediator variables with explanatory variables, but rather different explanatory variables, for example an environmental factor such as temperature (T) and D, where T and D ideally should be fully orthogonal to each other. In other words, you can compare T → EF and D → EF, but to look into the system that you have above you should use for example path analysis (SEMs) with arrows D → P, P → EF and D|P → EF, the latter indicating the effect of D on EF that is not mediated by P. Understanding (2), the authors do not find that P is more important than D in explaining EF, but that P provides a mechanistic explanation on the effect of D on EF. It is always like this in biodiversity research!

I have further major points that currently lead me to suggest rejection of this manuscript, but may hopefully allow the authors to write a new, more impactful manuscript.

(3) The EF measure in this study is not really invasion success but rather exclusion success or exclusion time, which could be calculated for each community as time until reaching the detection limit, using the three time points available. Especially for invaders, it seems more logical to start with low numbers, where they can be in full growth phase, than with carrying capacity. Still, I think the authors did a legitimate experiment by starting with carrying capacity, but then it is a test of exclusion success (better exclusion time) of the native community or persistence success (persistence time) of the invader.

(4) I would strongly suggest to not only use random forests for analysis. To make this study comparable to the earlier experimental study of the same system (Bell et al. 2005, 2009), it would be far better to use general linear (mixed) models. Regarding proxies for D, I would suggest to use more PCo axes instead of simply 6 derived groups of communities. For example, taking the first 5-10 PCo axes or so would seem to much better incorporate the community-composition aspect of D than do the 6 groups. Also, I'm surprised that the authors did not calculate some phylogenetic diversity measures, which should be possible because they did calculate phylogenetic distances to the invaders. Overall, I think a much richer analysis than presently done would be possible with the excellent data.

(5) It was difficult to find out the relevance of using two invaders and why it was an advantage that they were genetically closely related. The authors should mention more prominently that one represented an "external invader" and the other an "internal invader". Similar distinctions have been made for invasion in the Jena Experiment, for example. They should also more deeply discuss the effect of this "treatment factor".

---

## [Author Response]

[Editors’ note: the authors resubmitted a revised version of the paper for consideration. What follows is the authors’ response to the first round of review.]

Reviewer #1:I have some mixed feelings about this manuscript. On the one hand, I like the fact that natural microbial communities, rather than experimentally assembled communities or communities established through serial dilution, were used to study microbial invasions. Experimentally assembled microbial communities are much simpler than natural communities, and the use of serial dilution to generate diversity gradient runs the risk of confounding species diversity effects with species composition effects. The use of natural communities that differ in diversity and composition eliminates these problems.On the other hand, however, there are several issues in the manuscript that complicate the interpretation of the reported data.First, if I am not mistaken, it seems that no treatment was done to eliminate fungi in the water samples collected from tree holes, thus fungal communities were likely present in the experimental microcosms. If this is the case, fungi may have competed with bacteria for resources, and played a role in regulating bacterial invasion in the experiment. Thus although this study did not mention fungi at all, the reported findings may be vulnerable to alternative explanations related to the role of resident fungal communities for bacterial invasion.

Our apologies – there was indeed a treatment of cyclohexamide to remove fungi prior to producing the frozen archive of communities, but we neglected to mention it in the Methods and Materials before. They’re now in Field techniques: Field sampling of communities (line 452) and it’s also detailed in another publication of ours (Rivett and Bell, 2018).

Second, the use of only two closely related Pseudomonas species as invaders invites question about the generality of the reported findings. This small number of invading species contrasts sharply with the large number of different resident communities used in the experiment. A greater number of bacterial species, with different degrees of evolutionary relatedness, should have been used to ensure the robustness of the findings.

This is indeed a limitation of the study – in order to give the results wider generality, one would have to test this with a much more diverse set of invading species. However, this was not our primary research question and it was impractical to do this whilst also having good replication of the resident communities. For this reason we, like most similar experiments before us, chose to focus our experimental design on having a large number of resident communities, representing a gradient of diversity and composition. This was especially important in this study, because the diversity and composition differences amongst natural communities are far less extreme than those typically seen across communities artificially designed to differ in their diversity and composition (requiring more extensive sampling to capture differences). This was the first study of its type and so we feel this was appropriate.

We have now been more explicit about the reasoning for choosing two closely-related invaders in Methods: Laboratory techniques: Choice of invaders (lines 580-597).

Third, the choice of the large inoculum size (106 individuals per ml for each invader) is also puzzling. The initial population size of the invaders actually was so high that they were probably greater than population sizes of most, if not all, resident species. Such large propagule pressure almost never exists in nature, making this study an unrealistic exploration of biological invasions.

We agree that the inoculum of 10^5^ invader cells is large and may seem unusual from the standpoint of conventional invasion ecology. Reviewer 3 also notes that “Especially for invaders, it seems more logical to start with low numbers, where they can be in full growth phase, than with carrying capacity”. However, as noted by Mallon et al., (2018), “microbial invasions are usually an anomaly within the conventional invasion framework…natural or anthropogenic driven microbial invasions may start with an initially large population size” comparable to, or larger than that of the resident community. Mallon et al., (2018) gives the example of high levels of *E. coli* introduction via faecal deposition on soil, and in this study and previous studies similar to ours (e.g. van Elsas et al., 2012) they also introduced invaders at approximately the same density as the resident community. This idea is also supported by the ‘community coalescence’ paradigm in microbial ecology, which posits that microbes from distinct microbial communities frequently invade one another as entire communities (Rillig et al., 2015). We now give these examples in our Methods: Invasion assays (line 551) section, as well as providing specific estimates (based on previous studies) of invading and resident bacteria in natural tree hole communities. This system-specific evidence lends greater confidence to the idea that our invasion set-up – under which the number of invading bacteria were approximately equal to the number already established in the resident community – is not unrealistic. Furthermore, in this ecosystem where the main input is likely to be the annual autumn litterfall, natural invasions are likely to occur at low frequency but high magnitude (in terms of propagule/inoculum size).

This said, we agree that our use of the term ‘invasion success’ implied that we were talking about invasion-from-rare in the more conventional ecological sense, and have therefore changed the terminology to ‘invader survival’ for clarity, following van Elsas et al., (2012) and Mallon et al., (2018). This fits with the comment of Reviewer 3 on the use of a large inoculum that “Still, I think the authors did a legitimate experiment by starting with carrying capacity, but then it is a test of exclusion success (better exclusion time) of the native community or persistence success (persistence time) of the invader”.

In case of remaining doubt, other similar papers (e.g. De Roy et al., 2013; Eisenhauer et al., 2013) have introduced invaders at lower densities than the resident community (0.1, 1 and 5% of total starting biomass). Even under these conditions, the number of invading species is likely to be greater than the population size of many resident species – though the overall invading population size was less. However, perhaps more importantly, introducing a lower number of invaders did not appear to affect the way invasion resistance operates; invasion success is lower when they are introduced at a lower density (i.e. the intercept changes), but the diversity-invasion success relationship is broadly similar (i.e. the slope is the same) (De Roy et al., 2013).

In summary, there is evidence that our invasion conditions are comparable to previous studies, are a realistic approximation of high microbial invasion pressures in nature, and that the fundamental relationships of invasion resistance to composition do not appear to change with the number of individuals introduced. We therefore believe our choice of inoculum size was reasonable, balancing realism with practicality (e.g. ensuring the majority of invasions were detectable) and allowing us to study the community-side of invasion under conditions in which invader-side conditions were unlikely to be limiting.

Apparently, the two invader species were competitively inferior to some of the resident species, and declined over time in abundance after their introduction, and some invader populations were probably on their way to extinction. In this context, the term "invasion success", quantified as invader abundance, is also misleading.

We agree that the term “invasion success” – especially the “success” part, could be misleading. The invader did indeed decline from its starting density over time from its initial density, and so ‘success’ is probably the wrong term to use and we have now changed this to ‘*P. fluorescens/P. putida* invader survival’.

It is also worth emphasising here that the decline in abundance was almost inevitable; unless the invader completely took over the community, it could not expect to maintain the same density as the resident species. This is a typical result in microbial invasion experiments in closed (where no new resources are added) microcosms where the invader is introduced at a high density after an initial period of community establishment (we have now been up-front about this in the opening paragraph of the Results: Experimental set-up and general observations about invader survival, line 98). For example, van Elsas et al., (2012) introduced *E. coli* at 10^8^ cells/g soil and observed much more dramatic decline to a maximum of ~3000 (8 log CFU) invader cells/g after 24 hours, which declined further over time. Similarly, Eisenhauer et al., (2013) introduced *Pseudomonasputida* into established microcosms at 5% relative abundance, which declined to below 5% relative abundance in almost all communities. However, van Elsas et al., (2012) used the term ‘invader survival’ and Eisenhauer et al., (2013) used the term ‘invasion’, rather than ‘invasion success’ and so again, we agree that the term ‘success’ is misleading and have used ‘invader survival’ accordingly.

Overall, I feel these issues are significant and cannot be addressed with revision, and recommend the rejection of this manuscript for publication in eLife.Reviewer #2:In their paper, Jones et al. conducted an experiment testing whether residential community composition, species richness or productivity of several hundred bacterial communities determines the invasion success of invading bacterial species (two strains of Pseudomonas). The study is innovative, because the authors used natural bacterial communities isolated from tree holes (which they have used in previous studies, too).The found several community types that cluster according to species composition and two of types were particularly vulnerable to invasion, even though the species diversity was comparable to the other community types. This difference in community composition lead to differences in community productivity, which then in turn determined invasion success of the two invaders. The authors found that community diversity explained only negligible part of the invasion success. These findings have large implications for the field of microbial invasion ecology. The authors argue convincingly that future studies should go beyond the idea of community diversity and community composition by measuring as many community functions as possible and by conducting such experiment in a more dynamic system.The figures are great. They are carefully prepared, easy to understand and they can be interpreted without reading the full paper.

Thank you for these kind words recognising the innovation of our study and the communication of it – they were very motivating and helpful in rewriting the manuscript. In particular, regarding your kind comments on our figures – the figures have now changed as is necessitated by the rewrite, but we kept this comment in mind throughout and tried to use similar elements in our new figures to communicate the re-analysis as effectively as before (and hope we have done so!).

1. Impact statement: When reading through the abstract and the introduction, it seems to be that the statement should rather read "A microbial community's productivity is a better predictor of its invasion resistance than its diversity".

Abstract and Introduction have now been changed and should reflect this.

To me it seems that productivity is better than composition, but composition is still much better than diversity. For example, see statement on lines 106/107. This result is a bit lost in the title and impact statement. Or maybe the introduction should be targeted a bit more towards composition.

Yes this is the correct interpretation of our results. We agree that the diversity discussion is distracting, and so have now rewritten the Introduction to focus more on composition and its effects on productivity and thus invasion resistance.

2. L. 202: 48% is very high! Many researchers can only dream of such explanatory power.

Thank you for the reassurance – this gives us greater confidence that the results are important and sound.

3. L. 228: This is the only paragraph directly following up the Introduction, whereas the rest of the Discussion wasn't introduced as thoroughly. I definitely suggest rewriting the Introduction to move away from the pure diversity focus, bringing in composition, how composition and growth may be related, and maybe even the naturalisation hypothesis.

Thank you for your advice – we have rewritten the Introduction to lean away from diversity, and talk more explicitly about the relationship between composition and productivity.

4. L. 327-33: This is written in an unnecessarily convoluted and complex fashion. I suggest rephrasing to get the main message better across.

We agree and this has been removed/rewritten in the new manuscript.

Reviewer #3:As a plant ecologist and biodiversity scientist myself, I cannot judge some of the technical issues and therefore assess this manuscript from my own point of view. In fact, I think the authors did a very novel and excellent study in the still and ever-increasing, hot biodiversity-ecosystem functioning topic. But they do not take sufficient advantage of it (1) and suffer from a fundamental misconception (2).Regarding (1): experiments manipulating species richness have shown that reducing diversity in a working ecosystem generally reduces its functioning. In contrast, it is less clear if natural (i.e. working) ecosystems of higher species richness function better than natural ecosystems with lower richness. One of the main problems are confounding environmental variables such as soil fertility. Thus, the novel experiment that nobody had done so far would be to take the species of these natural ecosystems and recreate them under common environmental conditions, e.g. by planting plant communities in the same composition in an experimental garden as they occur in their natural environments. The study done by the authors is the first ever doing almost exactly this and thus super novel (I'm actually surprised myself that we never had the idea to do this novel experiment with plants!). I say "almost exactly", because it is not clear to me how well the authors could know that (a) their OTUs are species?

Thank you for your kind comments regarding the novelty of the study, and apologies that we did not communicate well the points about which you are unclear.

Regarding (a) OTUs do not equal species, though are the common species-like unit used in culture-independent microbial ecology using the sequencing method used here (16S amplicon sequencing) – sorry if you know this already I am just not presuming knowledge. In our bioinformatic pipeline, the 581 OTUs were assigned specieslevel taxonomy using BLASTn against a curated GreenGenes database (DeSantis et al., 2006). However, the species-level assignments are intended to be more descriptive than quantitative, given that species-level resolution for OTUs is unreliable (as we have now pointed out on lines 594-597 and 678-683). Nonetheless, despite these limitations we think that the use of OTUs is a reasonable decision for three primary reasons.

First, using OTUs makes our results comparable with previous microbial ecology studies looking at diversity-invasion relationships – which have tended to use at ‘strain diversity’ (sometimes attributing a species-level ID to these strains based on 16S amplicon sequencing of a monoculture). For example, De Roy et al., (2013) used 17 strains to make artificial resident communities, several of which were of the same genus. Similarly, van Elsas et al., (2012) randomly isolated strains from soil and used these to assemble communities (without sequencing the strains). Thus whilst some of the OTUs in our dataset that share the same genus may be the same species, previous studies with artificial communities also have this limitation and our results are at least comparable with them. We think that the use of OTUs provides an improvement upon experiments that use artificial communities constructed only from species that are able to be isolated, whilst maintaining a good level of comparability with them.

Second, the diversity-invasion relationship was intended to only be a sub-hypothesis in our study – though admittedly, our previous Introduction gave the impression that it was the main focus (and has now been rewritten in response to this issue being raised by Reviewer 2). In our new manuscript, we focus more on composition more widely and our use of functional groups rather than OTUs not only gives our models greater explanatory power but also means that we rely less on individual species-level assignments in interpreting our results.

Thirdly, our previous computational analyses of these communities shows that despite this lack of resolution, one can observe that there are distinct community classes that correspond to distinct functional profiles (inferred from predicted metagenomes), which we now mention in Discussion lines 314-317 and Methods: Dimensionality reduction approaches tested 700-702 (Pascual-García and Bell, 2020a).

And (b) their communities contained only those species, i.e. only bacteria and no other organisms (see also comment by second reviewer)?

Our apologies – as well as a filtering step to remove large organisms, there was indeed a treatment of cyclohexamide to remove fungi prior to producing the frozen archive of communities, but we neglected to mention it in the Methods and Materials before. They are now in Field techniques: Field sampling of communities (line 454-468) and it’s also detailed in another publication of ours using the same communities (Rivett and Bell, 2018).

There is still a possibility, however, that some organisms survived filtering and anti-fungal treatment (though we did not observe this in flow cytometry and/or visual inspection of the microcosms). Additionally, there may be bacteriophages present that are having an impact on invader survival, either directly or indirectly. Some of these limitations also apply to previous experiments – especially those that artificial communities created from diluted natural communities (He et al., 2014; van Elsas et al., 2012) – and so our results are at least comparable when talking about the effects of bacterial community composition. Nonetheless, it is certainly still possible that nonbacterial effects could explain some of the unexplained variation and we now discuss this in Discussion: Unexplained variation (line 387). We would very much like to see a future study that uses shotgun sequencing to unpick the relative contributions of different Kingdoms on invasion resistance in even less-altered natural communities.

Regarding (2): there have been long, often misguided discussions aboutthe importance of "species richness per se" in affecting ecosystem functioning vs. other factors, in particular species identities and species composition. The authors take this even further and compare the importance of diversity (including both richness and composition aspects, say D) with the importance of an "ecosystem function"(productivity proxies, say P) as covariate on invasion success, i.e. ecosystem functioning (say EF). This is like earlier arguments saying that when assessing effects of plant species richness (D) on community biomass (EF) one should consider only plots with high plant cover (P). However, the hypothesis is D → EF, and for that to be true it does not matter which mechanism (and how), in this case D → P → EF, mediates the causal relationship D → EF. When "importance" of explanatory variables is assessed, one cannot compare mediator variables with explanatory variables, but rather different explanatory variables, for example an environmental factor such as temperature (T) and D, where T and D ideally should be fully orthogonal to each other. In other words, you can compare T → EF and D →> EF, but to look into the system that you have above you should use for example path analysis (SEMs) with arrows D → P, P → EF and D|P → EF, the latter indicating the effect of D on EF that is not mediated by P. Understanding (2), the authors do not find that P is more important than D in explaining EF, but that P provides a mechanistic explanation on the effect of D on EF. It is always like this in biodiversity research!

This detailed critique was very welcome and fundamentally reshaped our re-analysis and discussion. We now split to analysis into 3 components:

– First, responding to Comment 4 of yours (below), we make sure that we are statistically capturing the effect of 16S composition in an appropriate way. To do this, we compared 7 types of random forests computing composition with 7 different dimensionality reduction approaches (see lines 709-722). Note that we substituted the single 6-level variable ‘community type’ (Pascual-García and Bell, 2020a) used in the previous manuscript with 19+1 variables representing the abundances of the main functional groups (Pascual-García and Bell, 2020b). Both of these groupings (the former of communities, the latter of OTU abundances) were generated using correlations amongst OTU abundances and there is congruity between the 19+1 functional groups and the previous 6 community types (Pascual-García and Bell, 2020b). Moreover, the 19+1 functional groups mean species’ abundances was found to be the most appropriate to explain invader survival, with a predictive power comparable to when all OTUs were included in the model (Supplementary Figure 1). Thus we opted to use the functional groups in this manuscript, which also represents a more conservative approach than our choice in the previous version, i.e. using 1 categorical variable with six levels/community types.

– Second, also in response to Comment 4 of yours, we now use random forest regression as only a part of our analysis – to identify the compositional and functional explanatory variables with the greatest explanatory power (which may overlap). We agree that one cannot really compare the importance of composition with the importance of productivity, since these co-vary and are not orthogonal. ‘Variable importance’ in random forests does not really mean ‘importance’ as it is in common usage, and in our previous manuscript we probably did not communicate this well and tended to conflate the two. Variable importance values in the random forest are estimates of each variables’ explanatory power independent of the other explanatory variables or samples in the model. Nonetheless, even if two explanatory variables have different importance values, the variance explained by them is likely to at least partly overlap with the variance explained by one or more other variables. Thus, we use the random forest to select compositional/genotypic and phenotypic variables whose overlap/mediation we then explore using SEMs (see below).

– Thirdly and finally, as recommended by you, we use structural equation models to estimate the extent to which productivity mediates the effect of starting composition on invasion resistance, using the subset of explanatory variables selected using random forest. We do this by building and comparing 3 SEMs assuming no, partial mediation and complete mediation of the effect of composition (as a latent variable including functional group abundances and diversity) by productivity (second latent variable including resident community cell density) on invasion (third latent variable including all 3 invader survival sampling points for both invaders). Comparing these models, we see that there is definitely mediation of the effect of composition by productivity, and the overall explanatory power (reflected in the R^2^ obtained for the Invasion latent variable) is similar whether you assume this is partial or complete. The fit of the partial model is the best of the three models (see AIC values), though this is sensitive to values below the strict detection limit of the assay and so we work with the more conservative scenario of strong mediation but with some independent effects in discussion. Note that the ‘partial mediation’ model includes the paths Composition → Productivity, Productivity → Invader survival and Composition|Productivity → Invader survival. Following your wording, this analysis concludes that productivity provides the partial, predominant mechanistic explanation for the effect of composition, but that there are likely also independent effects of composition.

I have further major points that currently lead me to suggest rejection of this manuscript, but may hopefully allow the authors to write a new, more impactful manuscript.(3) The EF measure in this study is not really invasion success but rather exclusion success or exclusion time, which could be calculated for each community as time until reaching the detection limit, using the three time points available. Especially for invaders, it seems more logical to start with low numbers, where they can be in full growth phase, than with carrying capacity. Still, I think the authors did a legitimate experiment by starting with carrying capacity, but then it is a test of exclusion success (better exclusion time) of the native community or persistence success (persistence time) of the invader.

Our response to Reviewer 1’s Third/Fourth point is relevant here, but in response to your specific comments – yes, we agree upon second consideration that our previous use of the term ‘invasion success’ was misleading. We now adopt the term ‘invader survival’ (akin, we think, to ‘exclusion success’ as you put it), which was used in a previous key manuscript exploring similar questions in artificial communities (van Elsas et al., 2012). We liked the suggestion of calculating the time until reaching the detection limit, though unfortunately our data was not suited to this. We only took 3 discrete measurements at the 3 timepoints; these timepoints were not extracted from a larger dataset with the sufficient longitudinal sampling (e.g. hourly reads), which would have made a calculation of exclusion time more feasible. Whilst preferable, therefore, a calculation of the time until reaching detection limit/exclusion time would have unfortunately been very imprecise and statistically underpowered.

(4) I would strongly suggest to not only use random forests for analysis. To make this study comparable to the earlier experimental study of the same system (Bell et al. 2005, 2009), it would be far better to use general linear (mixed) models.

As described above, we have taken this advice and random forest regression is now only one, ‘black box’ step in our analysis, which is complemented by the ‘clear box’, explicit SEM approach, as recommended by you. For further ease of comparison, we also now include a plot of diversity-invader survival with the linear model regression line and R^2^ included (Figure 3).

Regarding proxies for D, I would suggest to use more PCo axes instead of simply 6 derived groups of communities. For example, taking the first 5-10 PCo axes or so would seem to much better incorporate the community-composition aspect of D than do the 6 groups.

We agree that using the 6 derived groups is an imprecise summary of the effect of composition, and have taken your advice by moving away from these. In the first step of our analysis, we now compare seven different dimensionality reduction approaches for permuting the compositional explanatory variables, selecting the functional groups approach as the best in terms of explanatory power vs number of variables.

Also, I'm surprised that the authors did not calculate some phylogenetic diversity measures, which should be possible because they did calculate phylogenetic distances to the invaders. Overall, I think a much richer analysis than presently done would be possible with the excellent data.

We have now calculated Rao’s D (a phylogenetic equivalent of Simpson’s index) and included this in the random forests, which indicates no strong effect of phylogenetic diversity, as for phylogenetic distance. This is now shown in Figure 2 and 3 and we discuss it in our Results (lines 168-170), Discussion (lines 361-381) and Methods 659-670.

(5) It was difficult to find out the relevance of using two invaders and why it was an advantage that they were genetically closely related. The authors should mention more prominently that one represented an "external invader" and the other an "internal invader". Similar distinctions have been made for invasion in the Jena Experiment, for example. They should also more deeply discuss the effect of this "treatment factor".

See our response to Reviewer 2’s second comment. In short, we have followed your advice and now been more explicit about the reasoning for choosing two closely-related invaders in Discussion (lines 373-376) and Laboratory techniques: Choice of invaders (line 578-597).

References:

De Roy, K., Marzorati, M., Negroni, A., Thas, O., Balloi, A., Fava, F., Verstraete, W., Daffonchio, D., and Boon, N. (2013). Environmental conditions and community evenness determine the outcome of biological invasion. Nature Communications, 4(1), 1383. https://doi.org/10.1038/ncomms2392

Eisenhauer, N., Schulz, W., Scheu, S., and Jousset, A. (2013). Niche dimensionality links biodiversity and invasibility of microbial communities. Functional Ecology, 27(1), 282–288. https://doi.org/10.1111/j.13652435.2012.02060.x

He, X., McLean, J. S., Guo, L., Lux, R., and Shi, W. (2014). The social structure of microbial community involved in colonization resistance. The ISME Journal, 8(3), 564–574. https://doi.org/10.1038/ismej.2013.172

Mallon, C. A., Le Roux, X., van Doorn, G. S., Dini-Andreote, F., Poly, F., and Salles, J. F. (2018). The impact of failure: Unsuccessful bacterial invasions steer the soil microbial community away from the invader’s niche. The ISME Journal, 12(3), 728–741. https://doi.org/10.1038/s41396-017-0003-y

Pascual-García, A., and Bell, T. (2020a). Community-level signatures of ecological succession in natural bacterial communities. Nature Communications, 11(1), 2386. https://doi.org/10.1038/s41467-020-16011-3

Pascual-García, A., and Bell, T. (2020b). functionInk: An efficient method to detect functional groups in multidimensional networks reveals the hidden structure of ecological communities. Methods in Ecology and Evolution, 11(7), 804–817. https://doi.org/10.1111/2041-210X.13377

Rillig, M. C., Antonovics, J., Caruso, T., Lehmann, A., Powell, J. R., Veresoglou, S. D., and Verbruggen, E. (2015). Interchange of entire communities: Microbial community coalescence. Trends in Ecology and Evolution, 30(8), 470–476. https://doi.org/10.1016/j.tree.2015.06.004

Rivett, D. W., and Bell, T. (2018). Abundance determines the functional role of bacterial phylotypes in complex communities. Nature Microbiology, 3(7), 767–772. https://doi.org/10.1038/s41564-018-0180-0

van Elsas, J. D., Chiurazzi, M., Mallon, C. A., Elhottova, D., Kristufek, V., and Salles, J. F. (2012). Microbial diversity determines the invasion of soil by a bacterial pathogen. Proceedings of the National Academy of Sciences, 109(4), 1159–1164. https://doi.org/10.1073/pnas.1109326109